# Contrail formation within cirrus: ICON-LEM simulations of the impact of cirrus cloud properties on contrail formation

Pooja Verma[1,2] and Ulrike Burkhardt[1]

[1]Deutsches Zentrum für Luft- und Raumfahrt, Institut für Physik der Atmosphäre, Oberpfaffenhofen, Germany
[2]Formerly at Meteorologisches Institut, Ludwig-Maximilians-Universität, Munich, Germany

*Correspondence to*: Pooja Verma (Pooja.Verma@dlr.de) and Ulrike Burkhardt (Ulrike.Burkhardt@dlr.de)

**Abstract.** Contrail formation within natural cirrus introduces large perturbations in cirrus ice crystal number concentrations leading to modifications in cirrus microphysical and optical properties. The number of contrail ice crystals formed in an aircraft plume depends on the atmospheric state and aircraft and fuel properties. Our aim is to study contrail formation within cirrus and, in particular, the impact of pre-existing cirrus on the contrail formation processes. We modify the parameterizations for
contrail ice nucleation and the survival of contrail ice crystals within the vortex phase in order to consider the impact of the preexisting cirrus within the high-resolution ICON-LEM at a horizontal resolution of 625m over Germany. We then analyze the change in ice nucleation and survival due to the presence of cirrus ice crystals.

We have selected two different synoptic situations sampling a large range of cirrus cloud properties representative of average
thick in-situ formed cirrus and liquid origin, cirrus that are connected with a frontal system, and very thin cirrus within a high-pressure system. We find that contrail formation within cirrus mostly leads to increases in cirrus ice crystal numbers by a few orders of magnitude. Pre-existing cirrus has a discernible impact on contrail ice crystal number concentrations only if the cirrus ice water content and ice number concentrations are high or contrail ice nucleation rates are low. The sublimation of the cirrus ice crystals sucked into and subsequently sublimated within the aircraft's engine and the ice crystals mixed into the aircraft
plume experiencing sublimation and later deposition leads most of the time to an increase in the contrail formation threshold temperature which can be as large as 2K. This increase is large when the ambient ice saturation ratio is close to one and when the ice water content of the pre-existing cirrus is large. In areas of high ice water content, contrail ice nucleation rates can be significantly increased in particular at lower flight levels. After nucleation cirrus and contrail ice crystals compete for water vapor while the combined contrail and cirrus ice water mass grows. Once the aircraft plume gets trapped within the wake
vortices and descends the decrease in plume relative humidity leads to the sublimation of cirrus and contrail ice crystals. We find that this contrail ice crystal loss can be only seldomly slightly modified by the cirrus ice crystals. In particular, high concentrations of small ice crystals and large ice water content of the pre-existing cirrus cloud or low contrail ice crystal numbers can be sometimes associated with modest increases in the contrail ice crystal survival fractions. Cirrus ice crystals that get mixed into the plume can also lead to reduced ice nucleation and survival but negative changes are significantly
smaller.

# 1 Introduction

Cirrus clouds are very common in the upper troposphere and have a large impact on radiative transfer and, therefore, on climate and weather (Liou, 1986). Cirrus cool the atmosphere by reflecting incoming short-wave (solar) radiation and absorb and re-emit outgoing long-wave (terrestrial) radiation which warms the atmosphere. The size of both the short-wave and long-wave cloud radiative forcing depends on the macro- and microphysical cirrus properties (Ramanathan et al., 1989; Zhang et al., 1999). Aviation has a significant impact on upper tropospheric cirrus cloudiness (Boucher, 2013) due to the formation of contrails and due to aviation aerosol cloud interactions. Of the known aviation related radiative forcing components contrail cirrus is estimated to be the largest (Burkhardt and Kärcher, 2011) but the associated uncertainty is large (Lee et al., 2021). This is not unexpected since in IPCC style double $CO_2$ climate change simulations uncertainties in cloud responses are the main source of uncertainty in the equilibrium climate sensitivity (Stevens and Bony, 2013). In assessments of aviation related climate change (Lee et al., 2021) contrail cirrus and the indirect aerosol effects involving aviation aerosol emissions are the most notoriously difficult to estimate and the most uncertain (e.g. Righi et al., 2013, Kapadia et al., 2016, Lee et al., 2021) with uncertainties caused to a large degree by incomplete knowledge about number and ice nucleating properties of emitted and subsequently ageing aviation aerosols.

Contrail cirrus have been studied in great detail in observations (e.g. Gayet et al., 1996, Schröder et al., 1999, Voigt et al., 2017, Schumann et al., 2017) and in modelling. Modelling the life cycle of contrail cirrus, just as modelling natural clouds, involves processes on a large range of scales, comprising microphysical processes as well as large scale dynamics. Different approaches have been used, ranging from simulating single contrails over parts or the whole life cycle in LES (e.g. Lewellen et al., 2014, Unterstrasser, 2014, Paoli and Shariff, 2016) or NWP (Gruber et al., 2018) to simulating the evolution, properties and the climate impact of a large number of contrails in low resolution models with a significantly simplified microphysical treatment (Burkhardt and Kärcher, 2011; Bock and Burkhardt, 2016a, 2016b, 2019; Bier et al., 2017; Chen and Gettelman, 2013; Schumann et al., 2015). While LES is ideally suited to resolving the flow field around the airplane and, therefore, the contrail evolution in the first few minutes, numerical weather prediction and climate models are suited to simulating the contrail evolution which depends on the evolving atmospheric conditions controlled by synoptic scale variability.

Despite those efforts understanding contrail cirrus processes, many uncertainties connected with contrail cirrus radiative forcing remain stemming from uncertainties in the background upper tropospheric water budget and cirrus cloud properties, the contrail cirrus schemes and the impact of contrail cirrus on radiative transfer (Lee et al., 2021). Furthermore, the interaction between contrail cirrus and natural cirrus add to the uncertainty. Upper tropospheric natural cloudiness has been shown to decrease as a consequence of contrail formation and is, therefore, limiting the impact of contrail formation on climate (Burkhardt and Kärcher, 2011; Schumann et al., 2015; Bickel et al., 2020). The strength of this cloud adjustment is very uncertain. Furthermore, until now only contrail formation within a previously cloud-free air volume has been studied extensively. The impact of contrail formation within pre-existing clouds is largely unknown because it was thought to be secondary or even negligible. Contrail induced cloud perturbations within existing cirrus have recently been shown to lead to

changes in cloud optical depth that can be detected using satellite remote sensing (Tesche et al., 2016) which calls into question

the assumption that this effect is negligible.

Contrails form when relative humidity within the aircraft exhaust plume exceeds saturation relative to water as a consequence of the mixing of the plume air with ambient air (Schumann, 1996). Contrail formation is subject to the atmospheric state and aircraft and fuel parameters. The number of ice crystals nucleated during contrail formation depends on the thermodynamic state of the ambient atmosphere and on aircraft and fuel parameters, in particular the number of aerosol particles released by

the engine (Kärcher et al., 2015) but also on aerosol properties (Kärcher et al., 1998), their variability aerosol and inhomogeneities within the plume leading to ice crystals nucleating successively which has an impact on plume relative humidity and acts to decrease ice nucleation (Lewellen, 2020). At cruise altitude in the mid latitudes the atmospheric state is such that the number of emitted aerosol particles constrains the number of ice crystals forming within the contrail's jet phase (Bier and Burkhardt, 2019). At lower latitudes or altitudes this is not necessarily the case; here the thermodynamic state of the

ambient atmosphere, which is responsible for the evolution of relative humidity in the plume, often limits ice nucleation within contrails. Within the subsequent vortex phase, that lasts until a few minutes after emission, the aircraft induced wake vortices travel downwards and many of the contrail's ice crystals that are trapped within the vortices sublimate depending on the atmospheric state, aircraft parameters and the number of contrail ice crystals that nucleated within the jet phase (Unterstrasser, 2016).

Both, ice nucleation in the jet phase and ice crystal survival during the vortex phase may be modified by the existence of ice crystals from pre-existing clouds. Ice crystals from pre-existing cirrus that are sucked into the engine sublimate and lead to a small increase in the water vapor content of the plume (Gierens, 2012). Furthermore, cirrus ice crystals that are mixed into the plume after emission may sublimate as long as the plume is ice subsaturated (Kärcher et al., 1998). Once the plume is ice supersaturated water vapor may deposit on the cirrus ice crystals (Gierens, 2012). The resulting change in the plume water

vapor mixing ratio can lead to changes in the contrail formation threshold and in contrail ice nucleation. The cirrus ice crystals that are mixed into the aircraft plume compete with the newly nucleated contrail ice crystals for available water vapor and once trapped within the descending vortices both cirrus and contrail ice water sublimates. The competition of cirrus and contrail ice crystals for water vapor leads to a lower increase, or even a decrease, in the ice water mass associated with contrail ice than if no cirrus ice crystals were present which tends to decrease survival fractions. Within the descending vortices the sublimation

of the cirrus ice crystals increases the relative humidity and therefore limits the contrail ice sublimation as compared to the sublimation of the same sized contrail ice crystals disregarding the impact of the cirrus ice crystals. Both processes together may lead to a negative or positive change in the fraction of contrail ice crystals surviving the vortex phase. Any modification of the ice nucleation or survival during the vortex phase leading to changed ice crystal numbers after the vortex phase has an impact on contrail microphysical processes, contrail cirrus properties, optical depth and life time (Bier et al., 2017; Burkhardt

et al., 2018). On the one hand, increased ice crystal numbers lead to a stronger climate impact of contrail cirrus. On the other hand, changes in the contrail formation threshold can have large consequences in atmospheric conditions that are close to the

formation threshold and decide between a small aircraft induced sublimation of cirrus ice crystals and a large increase in ice crystal concentrations.

Kärcher et al. (1998) discussed the impact of pre-existing ice crystals that is sublimated in the combustor and that is mixed into the aircraft plume after emission on the formation of an observed warm (237K) contrail. They note that a large number of smaller cirrus ice crystals would be needed so that the sublimation of cirrus ice can increase relative humidity so much that a contrail could form. Gierens (2012) discusses the impact of cirrus ice sublimation during combustion and the impact of water vapor deposition on cirrus ice crystals mixed into the plume assuming average cirrus cloud properties and concludes that the effects are negligible.

We study here contrail formation within cirrus clouds as a first step towards evaluating the impact of cloud perturbations created by contrail formation within cirrus as observed by Tesche et al. (2016). We choose an approach in between LES and a global climate model, studying contrail formation in a numerical weather prediction setup at a resolution of a few hundred meters using parameterizations for contrail ice nucleation (Kärcher et al., 2015) and survival of contrail ice crystals in the vortex phase (Unterstrasser, 2016). We consider a wide range of cirrus cloud properties as simulated by the high-resolution ICON-LEM in weather forecasting mode (Heinze et al., 2017), calculating changes in the contrail formation criterion and in the number of ice crystals nucleating during contrail formation and in the contrail ice crystal loss during the vortex phase. In section 2 we introduce the ICON model and describe the contrail related processes that are part of our contrail scheme. The scheme consists of a parameterization for ice nucleation and for the ice crystal loss in the contrail's vortex phase and additions that consider the existence of pre-existing ice crystals from natural cirrus. We study contrail formation processes on two selected days that represent different synoptic situations over Germany and discuss the background natural cirrus cloud properties (Sect. 3.1). We analyze contrail ice nucleation within cirrus and the impact of pre-existing cirrus clouds on contrail formation threshold and ice nucleation (Sect. 3.2). In section 3.3 we estimate the fraction of contrail ice crystals surviving the vortex phase and analyze the impact of cirrus ice crystal on the vortex phase survival. We study the sensitivity of our results to different soot number emission indices and to a change in the upper tropospheric stability. Finally, in section 4 we present the changes in ice crystal number concentrations dependent on the cirrus ice crystal number concentrations without considering influences of aviation.

**2 Methods and Simulations**

We develop and implement a representation for contrail formation, based on parameterizations of ice nucleation in the jet phase and ice crystal loss during the contrail's vortex phase, in the ICON (ICOsahedral Non-hydrostatic) – LEM (Large-eddy model) (Zängl et al., 2014; Dipankar et al., 2015) that allows to study cirrus cloud modifications induced by contrail formation. We use a model set up that simulates the synoptic development over a limited domain, Germany, at a horizontal resolution of 625m using initial and boundary data coming from an operational NWP (Numerical weather prediction) system, COSMO (COnsortium for Small-scale MOdelling, Baldauf et al., 2011), at 2.8 km resolution. Instead of prescribing an air traffic

inventory, we prescribe air traffic everywhere in the upper troposphere studying the impact of pre-existing clouds on contrail formation, the contrail formation temperature threshold, ice nucleation and ice crystal loss in the vortex phase, for a large range of atmospheric states and cloud properties. We intentionally prescribe also air traffic at low altitudes down to about 7 km, that are usually not thought of as main air traffic levels, as air space over Germany has become very tight in the last years and more short distance flights have been moved to lower flight levels. Furthermore, vertical shifts in air traffic are being discussed in connection with the mitigation of aviation climate impacts (Fichter et al., 2005, Matthes et al., 2021).

## 2.1 ICON-LEM

ICON-LEM is based on the ICON (ICOsahedral Non-hydrostatic) modelling framework developed by the German Weather Service (DWD) and the Max-Planck Institute for Meteorology (Zängl et al., 2015, Dipankar et al., 2015). ICON solves a set of equations on an unstructured triangular grid based on successive refinement of a spherical icosahedron (Wan et al., 2013, Zängl et al., 2015). Time stepping is performed using a predictor-corrector scheme. A summary of the model configuration and a description of the physics package are given in Heinze et al., (2017) and references therein.

We use ICON in a LEM (large-eddy modelling) mode over Germany with realistic orography at a resolution of 625m and a time step of 3 seconds (Dipankar et al., 2015). The model has an option for 2 one-way nested domains. The model's high horizontal resolution combined with a vertical resolution of around 150m in the upper troposphere allows resolving relevant cloud processes, such as convection, while cloud microphysics, turbulence and radiation remain parameterized. Resolved cloud scale dynamics lead to improvements in structure and distribution of clouds and precipitation (Stevens et al., 2021). The heterogeneity in the cloud field and thus in the optical depth is largely resolved which enables a more realistic estimation of the radiative forcing relative to coarser resolution models. The model is initialized at 00 UTC from operational COSMO-DE analysis data (Baldauf et al., 2011) and relaxed at the lateral boundaries within a 20km nudging zone towards COSMO-DE analysis which are updated hourly. The initial and boundary condition data are interpolated to the ICON grids by using a radial basis function interpolation algorithm (Ruppert, 2007) and 3D variables are interpolated vertically during initialization. An evaluation of the model simulations has been presented by Heinze et al., (2017) and Stevens et al., (2020). The benefit of the high resolution of ICON-LEM or ICON-SRM (Storm Resolving Model) relative to lower resolution simulations was shown to lead to improvements in precipitation patterns, their location, propagation and diurnal cycle, and cloud properties, in particular the vertical structure and diurnal cycle (Stevens et al., 2020). In order to minimize computing time and disk space, we choose to run the model at 625m horizontal resolution. The benefit from increasing resolution from 625m to 156m was shown by Stevens et al., (2020) to be small.

### 2.1.1 Two moment cloud microphysics

The cloud microphysical scheme of ICON-LEM is based on Seifert and Beheng (2006) and includes microphysical processes in liquid, mixed phase and ice phase clouds. The microphysical two-moment scheme predicts mass mixing ratios and number

concentrations for six hydrometeors, cloud droplets, rain, ice, hail, snow, and graupel. The cloud cover scheme is an all-or-nothing scheme disregarding subgrid variability of total water. The microphysical scheme describes droplet formation and ice nucleation, growth and conversion processes between different hydrometeors, precipitation, and sedimentation. The parameterization for homogeneous and heterogeneous ice nucleation is based on Kärcher et al., (2006) and includes the competition between homogeneous and heterogeneous nucleation, and considers the impact of pre-existing ice crystals.

Heterogeneous nucleation is induced by INPs (Ice nucleating particles) with mineral dust concentrations prescribed according to Hande et al., (2015). Activation of INPs for heterogeneous nucleation is parameterized based on the simulation of the aerosol conditions with the COSMO MultiScale Chemistry Aerosol Transport (COSMO-MUSCAT) model (Wolke et al., 2004, 2012). A tracer is used to track the number of ice nuclei that have formed ice crystals and are therefore not available for ice nucleation anymore (Köhler and Seifert 2015).

## 170 2.2 Contrail scheme

We developed and implemented a contrail scheme within ICON-LEM, based on the parameterization of contrail ice nucleation (Kärcher et al., 2015) and ice crystal survival within the vortex phase (Unterstrasser, 2016), to study changes in cloud variables due to contrail formation within cirrus. Contrail formation, dependent on atmospheric and aircraft and fuel parameters, is calculated and contrail ice nucleation (Sect. 2.2.1) and ice crystal loss in the contrail's vortex phase (Sect. 2.2.3) is estimated.

We analyze contrail ice number concentrations after the contrail's vortex phase at a contrail age of ~5 minutes.

If contrails form within a pre-existing cirrus, the cirrus may have an impact on the contrail formation threshold, contrail ice nucleation and contrail ice crystal survival during the vortex phase depending on the cirrus macro- and microphysical properties. We consider the impact of the sublimation of natural cirrus ice crystals that are sucked into the combustor or mixed into the aircraft plume and lead to changes in water vapor mixing ratio in the young plume and changes in the contrail formation

threshold temperature and contrail ice nucleation (Sect. 2.2.2). After contrail ice nucleation, water vapor deposition on the contrail and cirrus ice crystals, that were mixed into the plume, leads to a relaxation of ice supersaturation. While shortly after nucleation ice supersaturation is usually so high that both contrail and cirrus ice crystals grow dependent on the size of their respective ice crystals, once ice supersaturation is close to saturation large cirrus ice crystals may grow at the cost of smaller contrail ice crystals (Lewellen, 2012). The competition of cirrus and contrail ice crystals for available water vapor leads to

smaller contrail ice crystals and, therefore, decreased contrail ice crystal survival fraction. The sublimation of ice crystals from the pre-existing cirrus, that are caught in the descending wake vortices, increases relative humidity within the subsaturated vortices and reduces the sublimation of contrail ice crystals. We analyze the combined impact of cirrus ice crystals during the diffusional growth phase and during vortex descent on the contrail ice crystal survival in Sect. 2.2.4.

In order to sample through a large number of atmospheric states with varying cloud properties without having to perform long

simulations we pick two different synoptic situations with very different background conditions and cloud properties. Both situations are part of one-day long ICON-LEM simulations described in Heinze et al., (2017). For each of those situations we

study contrail formation in the upper troposphere (above ~7 km) for only one timestep prescribing air traffic in each grid box of the simulation domain.

### 2.2.1 Parameterization of contrail formation and ice nucleation

Contrail formation depends on atmospheric conditions and fuel and aircraft dependent parameters and is described by the Schmidt-Appleman (SA)-criterion (Schumann, 1996). The temperature threshold for contrail formation depends on the slope of the mixing line, G, in a temperature-water vapor partial pressure diagram:

$$G = \frac{M_w c_p P_a}{0.622 Q (1-\eta)} ,$$ (1)

with $M_w$, $c_p$, $P_a$, $Q$ and $\eta$ are mass emission index of water vapor, specific heat capacity, atmospheric pressure, specific combustion heat and propulsion efficiency, respectively. We set the mass emission of water vapor to 1.25 kg (kg-fuel)$^{-1}$, specific combustion heat to 43.2 MJ (kg-fuel)$^{-1}$, and propulsion efficiency to 0.3 (Bock and Burkhardt, 2019). The temperature threshold of contrail formation, $T_{sa}$, is the ambient temperature for which the slope of the water saturation curve is equal to G, the slope of the plume mixing line. At ambient temperatures below that threshold, contrails will form if the ambient humidity is high enough. Contrails will only persist if ambient humidity is at least saturated relative to ice. At a given pressure level and for a given propulsion efficiency the slope of the mixing line depends on the ratio of emitted water vapor and combustion heat. An increase in water vapor emissions at constant combustion heat therefore leads to an increase in the slope of the mixing line and therefore to a higher temperature threshold of contrail formation.

Ice nucleation takes place within the first second after emission in the contrail's jet phase (Paoli and Shariff, 2016). The hot and moist air of the plume rapidly mixes with the cold and dry ambient air. If water saturation is exceeded within the plume, droplets form preferentially on emitted soot particles and background aerosols (Kärcher and Yu, 2009; Kärcher et al., 2015). The number of droplets that form within the contrail is dependent on the supersaturation and the size distribution and hygroscopicity of the aerosols. At current soot number emissions, volatile plume particles are generally too small to get activated. Once droplets have formed in the plume they rapidly freeze into ice particles by homogeneous freezing when plume temperatures fall below the freezing temperature. If the contrail formation threshold temperature is close to the ambient temperature then the maximum attainable plume supersaturation (when neglecting the decrease in supersaturation due to droplet formation) will be low and, therefore, only few soot particles will activate into water droplets and subsequently freeze (Kärcher and Yu, 2009, Kärcher et al., 2015). Close to the temperature threshold the apparent emission index (AEI$_i$) of contrail ice crystals increases rapidly with decreasing ambient temperature, $T_a$. Contrail formation close to the contrail formation threshold occurs often at low air traffic altitudes where air is relatively warm or in tropical or subtropical areas (Bier and Burkhardt, 2019). When contrails form far below the contrail formation threshold, AEI$_i$ is controlled by the soot number emission index. As ambient temperature decreases maximum attainable plume supersaturation increases and an increasing number of soot particles can activate and form ice crystals. The number of soot particles forming ice crystals is for temperatures

5K below the formation threshold close to the number of emitted soot particles (within approximately 25%). This means that in the extratropic at typical cruise levels ice crystal numbers in young contrails are mostly limited by the number of emitted

soot particles (Bier and Burkhardt, 2019).

We have implemented the parameterization of contrail ice nucleation based on Kärcher et al., (2015). The parameterization calculates the number of droplets that form and subsequently freeze within the contrail's jet phase. The number of droplets that form within the contrail is determined by calculating the number of droplets that can form at a given plume supersaturation and that lead to a decrease in relative humidity that balances the large increase in relative humidity due to the mixing of plume

and environmental air. All aerosols are assumed to activate and form droplets at the same time, $t_{act}$, called the "activation-relaxation time" neglecting the fact that aerosols that activate slightly earlier would have an impact on the plume relative humidity. This impact can sometimes be large in particular for large aerosol emissions (Lewellen, 2020).

We have calculated the apparent emission index of contrail ice crystals (AEI$_i$) prescribing a soot emission index (EI$_s$), assuming current day soot rich emissions of $2.5 \times 10^{15}$ soot particles per kg-fuel (Bräuer et al., 2021), on model levels between 7 km to

235 13 km altitudes. Figure 1 shows the dependency of AEI$_i$ on the difference between the ambient and the threshold temperature in the altitude range between 9.6 to 10.8 km for varying atmospheric conditions. Close to the formation threshold (T$_{sa}$-T$_a$ < 3K) AEI$_i$ rapidly increases with increasing difference between ambient temperature and the temperature threshold for contrail formation. At ambient temperature far below the temperature threshold a large percentage of the soot particles activate and form contrail ice crystals so that AEI$_i$ approaches EI$_s$. The apparent emission index of ice varies for fixed difference between

240 ambient and Schmidt-Appleman temperature since atmospheric conditions, i.e. pressure, water vapor mixing ratio and the Schmidt-Appleman temperature, are not constant.

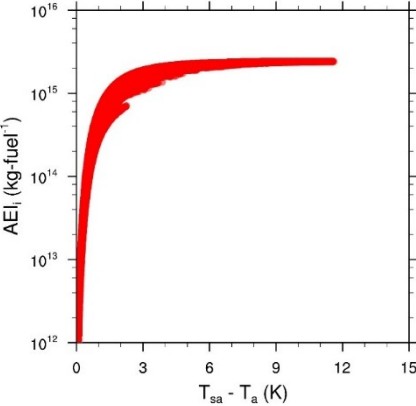

**Figure 1: Apparent Emission Index, AEI$_i$ versus the difference between ambient temperature, T$_a$, and Schmidt-Appleman temperature, T$_{sa}$, for the 26$^{th}$ April 2013 at altitudes between 9.6km to 10.8km for various combinations of atmospheric pressure, ice saturation ratio and temperature threshold for contrail formation.**

## 2.2.2 Impact of natural cirrus on contrail formation and ice nucleation

The sublimation of cirrus ice crystals within the engine acts to increase water vapor emissions coming from fuel combustion and therefore changes the slope of the plume mixing line. Cirrus ice crystals that are mixed into the aircraft plume before nucleation (within the first second of the plume life time) sublimate partially as long as the plume is ice subsaturated and increase the plume water vapor mixing ratio. Once the plume is ice supersaturated deposition on the cirrus ice crystals will take place, decreasing the water vapor mixing ration. Assuming a mass-based air to fuel mixing factor at engine outlet, $N_0$, of 70 kg-air kg-fuel$^{-1}$, we estimate the change in the plume's water vapor content per kg-fuel due to the presence of cirrus ice crystals (with a mean mass 'm') at the time of aerosol activation ($t_{act}$), $M_{cir}$ [kg kg-fuel$^{-1}$]:

$$M_{cir} = (q_{ci} * N_0) + \int_{t_0}^{t_{act}} \left( \left(\frac{dm}{dt}\right)_{sub} - \left(\frac{dm}{dt}\right)_{dep} \right) n_{ent} \, dt \qquad (2)$$

with $q_{ci}$ the ice water mass mixing ratio of cirrus [kg kg-air$^{-1}$], $n_{ent}$ is the apparent number emission index for cirrus ice crystals entrained into the plume from the surrounding air [kg-fuel$^{-1}$] which increases with plume dilution. $n_{ent}$ and the dilution are calculated using equation 18 and 12 of Kärcher et al. (2015). The sublimated water mass per cirrus ice crystal, $\left(\frac{dm}{dt}\right)_{sub}$, and deposited water mass on a cirrus ice crystal, $\left(\frac{dm}{dt}\right)_{dep}$, and their time integrated values are estimated as explained in Appendix A. The first term of equation 2 describes the cirrus ice water that is sublimated within the engine per kg fuel burned. The second term stands for the time integrated change in water vapor mixing ratio due to sublimation and deposition on cirrus ice crystals that are mixed into the plume after emission ($t_0$) and before aerosol activation ($t_{act}$), assuming that we can neglect any changes in the time of aerosol activation that may be caused by the presence of cirrus ice crystals. The impact of cirrus ice crystals on the aviation induced plume water vapor content is discussed in section 3.2.

Sublimation and deposition on cirrus ice crystals lead to a deviation of the plume's water partial pressure away from the mixing line approximation, with variations largest shortly after emissions due to the plume's large subsaturation and high temperature. We approximate the evolution of the plume properties by a new mixing line, treating the change in the plume's water vapor content from the sublimation and deposition as a change in the water vapor emission. When calculating the slope of the new mixing line we add the $M_{cir}$ to the mass emission index of water vapor, $M_w$. The new slope for the mixing line $G_{ci}$ is:

$$G_{ci} = \frac{(M_w + M_{cir}) c_p P_a}{0.622 Q (1-\eta)} \qquad (3)$$

When calculating the slope of the time series of water partial pressure at the time of contrail ice nucleation we find deviations from the simple mixing line approximation (equation 3) of a few tenth of a percent. Only when assuming a very large cirrus ice crystal number concentration of $5*10^6$ m$^{-3}$ the deviation from the mixing line slope at the time of aerosol activation can reach values of up to 1%. This agrees with Gierens (2012) who finds that for a plume age of 1 second, at typical cirrus ice

crystal concentrations and typical atmospheric conditions, the deposition time scale is 2 to 3 orders of magnitude smaller than the dynamic jet timescale, which indicates that cirrus ice crystals grow too slowly to effectively reduce plume supersaturation production due to cooling.

If the slope of the mixing line, $G_{ci}$, is larger than the slope of the mixing line, that neglects the impact on the background cirrus ice crystals then the temperature threshold for contrail formation is increased. Plume supersaturation can occur earlier and the maximum attainable relative humidity, that is reached within the plume when neglecting the decrease in supersaturation due to aerosol activation and droplet formation, can be larger. Therefore, ice nucleation is increased.

### 2.2.3 Parameterization of ice crystal loss during vortex descent

After contrail ice nucleation the plume ice water mass grows and approximately ten seconds after the emission, the exhaust plume including the newly formed ice crystals gets trapped in a pair of counter rotating vortices (primary wake) that are created when the vorticity sheet originating from the pressure differences at the aircraft wings rolls up (Paoli and Shariff, 2016). The counter rotating vortices propagate downward depending on atmospheric stability and aircraft properties, such as weight, wing span and speed (Gerz et al., 1998). The density contrast between the air in the vortex, that descends through a stably stratified

atmosphere, and the surrounding creates vorticity that is shed upwards (secondary wake) and part of the exhaust, between 10% and 30% (Gerz et al., 1998), are detrained into the secondary wake. The secondary wake stays close to the flight level. The primary wake often descends a few hundred meters. Many ice crystals within the primary downward propagating vortices sublimate due to adiabatic heating and the associated decrease in relative humidity, while the ice crystals in the secondary wake are more likely to survive. Survival of the ice crystals in the vortex regime depends on atmospheric temperature,

humidity, the number of nucleated ice crystals and the maximum vertical displacement of the vortices. After vortex descend most of the air that was forced downwards rises again creating a vertically extended contrail.

The parameterization for the impact of the vortex descent on contrail properties in ICON-LEM is based on the work of Unterstrasser (2016). He used LES to study for a number of different aircraft (with differences in weight and wing span) and varying conditions of the surrounding atmosphere, the vertical extent of the contrail and the survival fraction of ice crystals.

The parameterization estimates (1) the maximum vertical displacement of the vortices in the atmosphere, (2) the vertical extent of the contrail which is given by the maximum vertical displacement of the vortices if ice crystals survive at the location of maximum displacement and smaller otherwise and (3) the survival fraction of the contrail ice crystals caused by the change in the relative humidity connected with adiabatic warming of air due to vortex descent. The parameterization captures the dependence of the survival fraction on ice supersaturation, temperature, contrail ice crystal sizes and atmospheric stability. We

use the parameterization assuming aircraft properties of medium sized aircraft (Aircraft type A350 or B767) (Unterstrasser,2016 table 1) to estimate the survival fraction of ice crystals and the vertical extent of the contrail after vortex descend. The contrail cross sectional area is given by the contrail vertical extent times the aircraft's wing span.

The survival fraction of nucleated contrail ice crystals is defined as:

$$Survival\ fraction = \frac{number\ of\ contrail\ ice\ crystal\ surviving\ vortex\ descent}{total\ number\ of\ nucleated\ contrail\ ice\ crystal} \qquad (4)$$

A survival fraction of one means that all ice crystals survive the vortex descent and zero means all nucleated contrail ice crystals sublimate.

### 2.2.4 Impact of cirrus ice crystals on growth of contrail ice crystals after nucleation and on sublimation of contrail ice crystals during vortex descent

When contrails form within cirrus the cirrus ice crystals entrained into the plume can have an impact on the growths of contrail
ice crystals after nucleation and on the subsequent loss of contrail ice crystals during the contrail's vortex descent. Cirrus and contrail ice crystals together act to relax the plume's supersaturation towards the ice saturation value. We use the diffusional growth equation (e.g. Pruppacher and Klett, 1997, Paoli and Shariff, 2016, Lewellen, 2012, Appendix A) in order to estimate the temporal evolution of water deposition on the contrail and cirrus ice crystals for 9s after nucleation before vortices start to descend. Once the plume's relative humidity approaches ice saturation the smaller contrail ice crystals may sublimate while
larger cirrus ice crystals may still grow due to the dependence of the saturation vapor pressure on the curvature of the ice crystals (Kelvin effect). In this situation the difference in ice crystal sizes between cirrus and contrail ice crystals increases. This behavior may be often found in contrails (Lewellen, 2012) and is largest in areas of low background relative humidity and large cirrus ice crystals. Since the presence of cirrus ice crystals limits the deposition on the contrail ice crystals, more contrail ice crystals may be lost during vortex descent since their associated ice mass is lower.
When the plume gets trapped in the wake vortices and the vortices propagate downward, temperature increases and relative humidity decreases, causing contrail and cirrus ice crystals to sublimate as soon as air becomes subsaturated. The sublimation of both the cirrus and the contrail ice crystals moisten the air volume of the vortex. Therefore, the sublimation of cirrus ice crystals reduces the sublimation fraction of the contrail ice crystals by weakening the decrease in relative humidity within the vortex. We estimate how much cirrus ice water sublimates in the time it takes to completely sublimate the contrail ice water
within the descending vortices using the diffusional growth equation, assuming spherical particles, and setting the saturation ratio to the slightly sub-saturated value of 0.98, a value that can be typically found in the descending vortices (personal communication Simon Unterstrasser), resulting from the decrease in relative humidity due to the temperature increase and from the moistening due to sublimation. We roughly estimate the ratio of sublimated cirrus water mass, $M_{cirrus}$, and sublimated contrail water mass, $M_{contrail}$, dividing the diffusional growth equation for cirrus ice crystals by the one for contrail ice crystals
and multiplying with the ratio of the number of ice crystals within the air volume:

$$\frac{dM_{cirrus}}{dM_{contrail}} \simeq \frac{dm_{cirrus}/dt}{dm_{contrail}/dt} * \frac{N_{cirrus}}{N_{contrail}} \qquad (5)$$

Once we have roughly estimated the amount of cirrus ice water that sublimates during vortex descent while contrail ice water sublimates, we calculate the impact of the sublimated cirrus ice water on contrail ice crystal sublimation. We proceed in the following way: a. Estimate diffusional growth on contrail and entrained cirrus ice crystals before vortex descent. b. Estimate the cirrus ice water mass that sublimates within the time that contrail ice crystals sublimate which is either given by the time the vortex descends or by the time it takes to sublimate all contrail ice crystals. c. Adjust the plume relative humidity in the parameterization Unterstrasser (2016) consistent with the sum of water deposition on the cirrus ice crystals during the growth phase and the cirrus ice water mass sublimated during vortex descent. d. We recalculate the number of contrail ice crystals that sublimate and the fraction that survives the vortex descent.

In section 3.3 we will discuss the impact of cirrus ice crystals on the loss of contrail ice crystals during the vortex phase comparing to the ice crystal loss that the contrail would have experienced in exactly the same situation except for the absence of cirrus ice crystals.

## 2.3 Simulations - analysis

We study contrail formation in a large variety of atmospheric states and cloud properties over Germany using ICON-LEM at a horizontal resolution of 625m and a vertical resolution of approx.150m. In order to sample many different atmospheric conditions, we prescribe air traffic within each grid box at altitudes of between 7km and 13km assuming an average fuel consumption of 6 kg-fuel/km which is typical for cruise conditions over Germany according to the Aviation Environmental Design Tool AEDT air traffic inventory (Wilkerson et al., 2010). Soot number emissions are set to $2.5 \times 10^{15}$ kg-fuel$^{-1}$ in line with Bräuer et al., (2021). We study two different synoptic situations, the 24[th] April 2013 6am and the 26[th] April 2013 5pm (Sect. 3.1) starting our model with output from longer simulations with ICON-LEM that started on the respective days at midnight (Heinze et al., 2017). The success of the model simulating the large-scale synoptic situation and the associated cloud fields of those days is documented in Heinze et al., (2017). We calculate contrail ice nucleation within cirrus and the subsequent ice crystal loss in the vortex phase on the 24[th] April 6am in approximately 6 million cloudy grid boxes and on the 26[th] April 2013 5pm in approximately 5.5 million cloudy grid boxes between 7km and 13km with cloudy grid boxes defined as ice water content (IWC) > $10^{-11}$ kgm$^{-3}$. For the calculation of ice crystal loss in the vortex phase we assume a fixed Brunt-Väisälä frequency of 0.012 s$^{-1}$ and calculate the sensitivity to the assumed stability. Assuming a fixed Brunt-Väisälä frequency reduces the degrees of freedom in our calculations making it easier to isolate the impact of contrail formation on cirrus properties. When exploring the sensitivity of our results to the stability we assume a Brunt-Väisälä frequency of 0.005 s$^{-1}$. We furthermore study the sensitivity of the contrail ice crystal survival fraction on the soot number emissions, reducing the emission index of soot by up to 80%.

# 3 Impact of pre-existing cirrus on young contrail properties

We perform case studies for two different synoptic situations, a high-pressure system over central Europe on the 24[th] April and a frontal passage on the 26[th] April 2013. In section 3.1 we introduce the synoptic situation and the cirrus properties found at that time over Germany. We study contrail formation, contrail ice nucleation and survival and the impact of the pre-existing cirrus on contrail formation and ice nucleation (Sect. 3.2) and on the ice crystal loss in the vortex phase (Sect. 3.3).

## 3.1 Synoptic condition

We selected two days for our analysis, the 24[th] April and 26[th] April 2013. The days were part of the HOPE measurement campaign (Macke et al., 2017) that had the goal of evaluating the performance of the high-resolution ICON simulations. The synoptic situation on those two days was very different which allows us to study contrail formation within pre-existing cirrus in strongly varying synoptic settings leading to distinct cloud microphysical properties. On the 24[th] April a high-pressure system dominated over Germany with close to clear sky conditions in many areas and some thin in-situ formed cirrus. The 26[th] April saw a passage of a cold front over Germany moving towards the southeast connected with a conveyor belt that was supplying the upper troposphere with moist air. Cloudiness was rapidly increasing and strong frontal convection, geometrically thick clouds and precipitation could be found along the front. Most of the cirrus clouds on this day had properties typical for average thick in-situ formed cirrus and a small fraction of the cirrus had properties that are typical for liquid origin cirrus (Krämer et al., 2020).

The simulations for those days were part of the model evaluation performed by Heinze et al., (2017) and Stevens et al., (2020). Heinze et al., (2017) showed that the synoptic systems on those days were simulated well by ICON. The high resolution of the ICON-LEM simulations led to improvements e.g. in the vertical cloud structure and the diurnal cycle of clouds (Stevens et al., 2020). On the 24[th] April cloudiness in general may be overestimated in comparison with MODerate Resolution Imaging Spectroradiometer MODIS images over central Germany while cirrus clouds, for instance in the northwest of Germany, are largely missed or are too thin in the simulations. Over the middle of Germany, a large thin cirrus cloud field with low ice water content and ice crystal number concentration is simulated in an ice saturated environment and persists for several hours. The cirrus field is spatially very homogeneous. On the 26[th] April, ICON simulates the frontal passage realistically and shows a slight underestimation of cloud fraction, with a good agreement regarding the cloud water path (Heinze et al., 2017). The cirrus is scattered and microphysical properties of the cirrus vary significantly. Lifting within the frontal zone ensures a continues water vapor supply in the upper troposphere and provides ice supersaturated conditions within the relatively thick cirrus layer. The conditions are therefore favourable for contrail formation and ice crystal growth.

We have performed a CFAD (Cloud Frequency Altitude Diagram) analysis to examine the properties, in particular the ice crystal number concentration, the mean volume diameter of ice crystals and ice water content (IWC), of the cirrus clouds (Fig. 2). The CFAD diagram provides information about the frequency of occurrence (probability density) of the cloud properties at different atmospheric temperatures. Figure 2 shows the frequency of occurrence of ice crystal number concentration (Fig. 2

b,e), the mean diameter of ice crystals (Fig. 2 c,f) and ice water content IWC (Fig. 2 a,d) at different temperatures in the cirrus cloud field over Germany for the 24[th] and 26[th] April 2013 at 6-7 am and 5-6 pm, respectively. On the morning of the 24th April the cirrus cloud over Germany is relatively homogeneous and the probability of cloudy areas reaching ice crystal number concentrations of roughly $10^5$ m$^{-3}$ at about 220K, a typical cruise level, is 0.01%. At the same time IWC is low and only in 0.01% of the cirrus at 220K values of $3*10^{-3}$gm$^{-3}$ are reached. On the evening of the 26[th] April, the distributions of ice crystal number concentration and IWC are much wider with 0.01% of cloudy areas reaching values of up to $10^8$ m$^{-3}$ and 0.5gm$^{-3}$ at 220K. Describing the width of the distribution by the values occurring with a probability of 0.01%, the diameter of the ice crystals varies strongly with temperature and ranges between 20µm and 200µm at temperature 210K and between 20µm and 400µm at temperature 230K on the 24[th] April 2013 and on the evening of the 26th April 2013 between less than 10µm and 200µm and between 15 µm and 600µm at temperature 210K and 230K, respectively. The most striking difference between the cirrus properties on the two days are the large differences in ice number concentrations with extrema in ice number concentrations at 220K about 3 orders of magnitude higher on the 26[th] April than on the 24[th] April. At the same time extrema in IWC are about 1 order of magnitude larger on the 26[th] April and the probability of low ice crystal sizes is increased. Those high concentrations of small ice crystals on the 26[th] April are likely connected with homogeneous freezing events happening in the areas of high ice supersaturation caused by lifting in the conveyor belts and with the freezing of droplets lifted within convective systems along the front. The vertical line in the diameter diagram (Fig. 2c and f) is an artefact coming from the lateral boundary conditions supplied by COSMO which is run using a 1-moment microphysical scheme. When using COSMO data for the forcing fields, a diameter of 100µm and associated ice crystal numbers are assumed (personal communication Axel Seifert, DWD) leading to an increased probability of ice crystals sizes of 100µm particularly in areas close to the model edge.

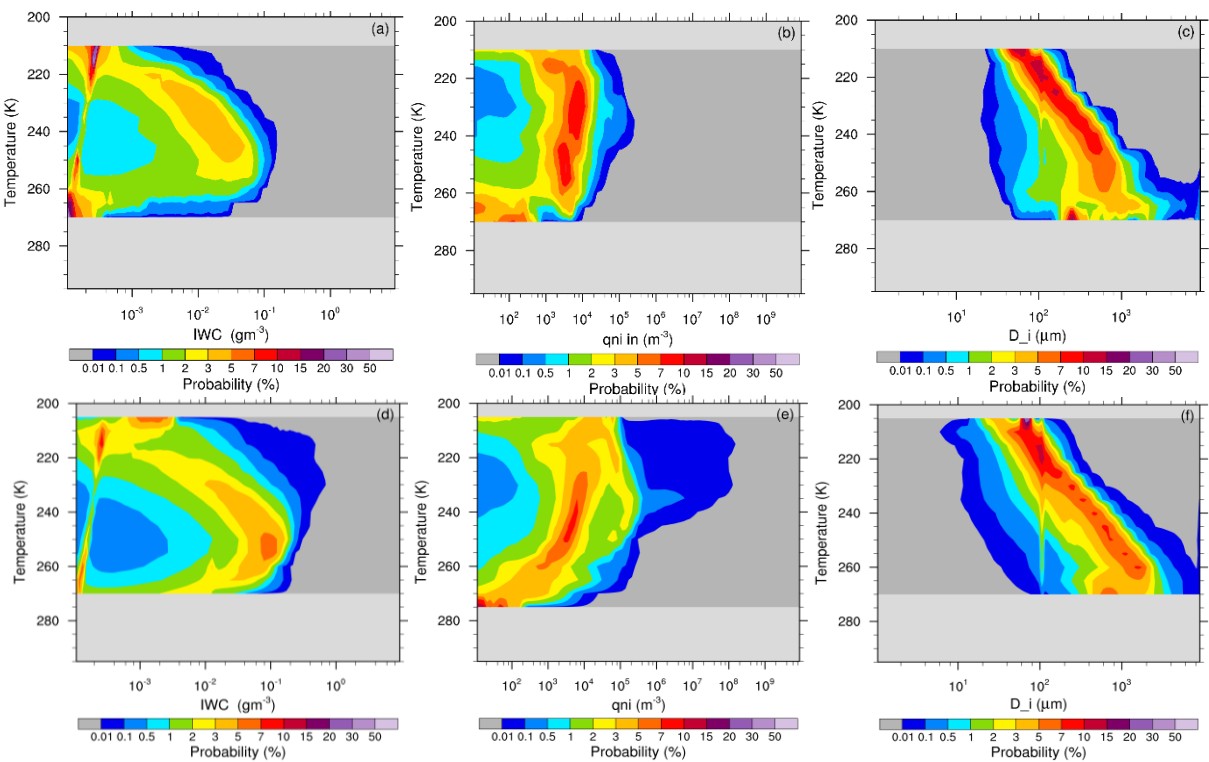

**Figure 2: Frequency of occurrence of IWC (a, d), ice crystal number concentration (b, e), and mean volume diameter of ice crystals (c, f) on the (a,b,c) 24th April 2013 at 6 - 7 am and (d,e,f) 26th April at 5 – 6 pm. The frequencies of occurrence refer to individual temperature bins.**

## 3.2 Impact of the pre-existing cirrus on contrail formation and ice nucleation

We study the impact of cirrus ice crystals, that are either sublimated within the combustor or mixed into the plume before ice nucleation, on the contrail formation threshold temperature and on ice nucleation. When aircraft fly through a cirrus cloud, air together with ice crystals get sucked through the engine inlet and sublimate. Furthermore, cirrus ice crystals get mixed into the plume causing sublimation or deposition depending on the plume's ice saturation ratio. The presence of cirrus ice crystals leads to a change in the total water vapor in the exhaust plume, usually increasing the water vapor content of the plume. The

increase is largest when the cirrus IWC is large. In the following we will call the sum of the change in the water vapor content caused by the sublimation of cirrus ice crystals and by the deposition on the cirrus ice crystals while the plume is ice supersaturated together with the water vapor emissions due to the combustion of fuel the 'aviation induced increase in water vapor'. We estimate the increase in the aviation induced increase in water vapor for cirrus clouds that have properties as displayed in figure 2. The ratio of the change in plume water vapor concentration due to sublimation of or deposition on cirrus

ice crystals and the aviation induced increase in water vapor is mostly small. The change connected with the presence of cirrus

ice crystals contributes often only a few thousands to a few hundreds of a percent to the aviation induced increase in water vapor (Fig. 3) in line with the fact that we consider clouds with as little as $10^{-11}$ kgm$^{-3}$. Maximum contributions reach values of half a percent on the 24$^{th}$ April 6am (probability of $10^{-4}$) and of 10% on the 26$^{th}$April 5pm (probability of $5*10^{-2}$) and up to nearly 20% at a probability of $10^{-2}$. The PDF of the changes only due to sublimation of cirrus ice crystals in the combustor is simply shifted towards lower values. This is despite the fact that the change in the plume water vapor concentration due to the sublimation of and deposition on cirrus ice crystals that were mixed into the plume can also be negative.

The change due to the sublimation of cirrus ice within the combustor is roughly in agreement with the cirrus ice water content reaching values of 0.5 gm$^{-3}$ at 220K (Fig. 2d). Assuming a pressure of 230 hPa the ice water mass mixing ratio can be estimated and prescribing an air to fuel mixing factor of 70 kg-air/kg-fuel the cirrus ice water mass sublimated in the engine per mass of fuel burned can be shown to agree with the ratio of sublimated cirrus ice water mass and aviation induced increase in water vapor (Fig. 3).

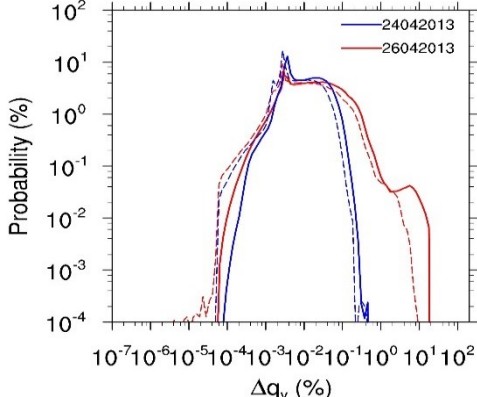

**Figure 3: Probability distribution of the ratio of the change in water vapor concentration due to the impact of cirrus ice crystals and the aviation induced increase in the water vapor '$\Delta q_v$' at aerosol activation, roughly 1 second after emission, for the 24$^{th}$ 6am (blue) and 26$^{th}$ April 2013 5pm (red) for areas with temperatures lower than 233.15K, ice saturation ratio larger than 1 and IWC larger than $10^{-11}$ kgm$^{-3}$. The water vapor emission index is set to 1.25. Cloud properties for the two days are displayed in figure 2. Dashed lines indicate the change in the aviation induced increase in water vapor due to the sublimation of cirrus ice crystals in the combustor only and solid lines indicate the change when also accounting for the sublimation of cirrus ice mixed into the plume and the deposition on cirrus ice once the plume is ice supersaturated.**

### Temperature threshold for contrail formation

Even though the change in the water vapor concentration due to sublimation of/deposition on cirrus ice crystals has often only a small impact on the aviation induced water vapor increase, it can result in a significant change of the Schmidt-Appleman threshold temperature, $T_{sa}$, (Fig. 4 c,e). On the 26th April 2013 temperatures are usually between 4 K and 10 K and between 1.5 K and 7 K lower than the contrail formation threshold (Fig. 4 a,b) on the main flight levels between 10.3 km and 10.8 km and between 9.6km and 9.8km , respectively. On the 24$^{th}$, at height levels between 9.6km and 9.8km i.e. at a pressure of around 270 to 280 hPa, temperatures lie mostly up to 5k below the Schmidt-Appleman threshold temperature. The change in $T_{sa}$ on the 24$^{th}$ is always very low, ranging between 0.04 K and -0.01 K, (Fig. 4e) consistent with the small impact of cirrus ice crystals

on the aviation induced water vapor increase (Fig. 3). On the 26[th] April, the change in the threshold temperature is often low but changes in $T_{sa}$ can exceed values of 1.6 K in the lower and warmer atmospheric levels (between 9.6 km and 9.8 km at ambient temperatures between 223 K and 227 K) (Fig. 4d) and 2K at higher atmospheric levels (between 10.3 km and 10.8 km at ambient temperatures between 215 K and 221 K) (Fig. 4c). Negative changes of the contrail formation threshold can occur but are small and relatively seldom. On the 26[th] April, large changes in the contrail formation threshold temperature are mainly associated with low ambient relative humidity (Fig. 4 c,d). An ice saturation ratio of 1 within a cirrus cloud is often indicative of a large ice crystal density that leads to an efficient relaxation of relative humidity to the saturation value. It is exactly in those areas that changes in $T_{sa}$ are high. The high saturation ratios i.e. ice saturation ratios of 1.4 and 1.3, on the other hand, indicate low ice crystal concentrations and ice water content and are likely to be the areas in which homogeneous and/or heterogeneous nucleation may occur within the next few time steps. In areas of high ice saturation ratio, the change in $T_{sa}$ is negligible. In the following we will explore the reasons for large changes in $T_{sa}$ in more detail.

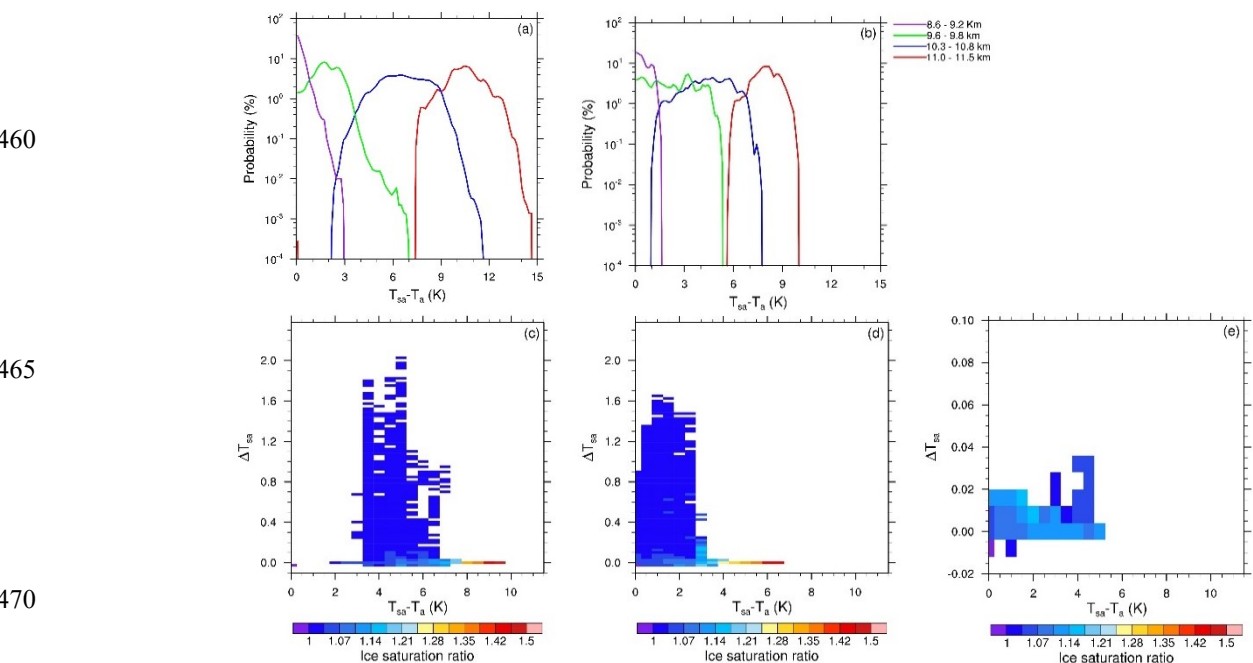

**Figure 4: Probability distribution of the difference between the Schmidt-Appleman temperature threshold of contrail formation ($T_{sa}$) and the ambient temperature ($T_a$) at 11 – 11.5 km (red), 10.3 – 10.8 km (blue), 9.6 – 9.8 km (green) and 8.6 – 9.2 km (purple) on the 26[th] April (a) and 24[th] April (b). Difference between the Schmidt-Appleman temperature threshold ($T_{sa}$) and the ambient temperature and change in $T_{sa}$ due to impact of cirrus ice crystals on the 26[th] April 2013 5pm at altitudes of 10.3 (~250 hPa) to 10.8km (~225 hPa) (c) and at 9.6 (~280 hPa) to 9.8km (~270 hPa) (d) and on 24[th] April 2013 6am at altitudes of 9.6 (~280 hPa) to 9.8km (~270 hPa) (e). In all figures the difference of ambient air temperature and $T_{sa}$ refers to the $T_{sa}$ that is not modified due to the sublimation of pre-existing cirrus ice.**

**Contrail ice nucleation**

Large differences between the ambient temperature and the threshold temperature for contrail formation lead to high contrail ice nucleation rates (Fig. 1). Contrail ice nucleation within pre-existing cirrus leads to large perturbations in the ice crystal number concentration of the cirrus cloud field. On the 24th April 2013 cirrus ice crystal number concentrations at 220K reach values of about $10^5$ m$^{-3}$ with a probability of 0.01% (Fig. 2b) while contrail ice nucleation leads to ice crystal number concentrations of between $10^7$–$10^8$m$^{-3}$ (Fig. 5b). On the 26th April 2013 the frontal system and the associated large moisture transport into the upper troposphere leads to localized nucleation events so that cirrus ice crystal number concentrations of up to $10^8$ m$^{-3}$, the same order of magnitude as the contrail perturbations (Fig. 5a), occur with a probability of 0.01% (Fig. 2e). This means that approximately 5 minutes after contrail ice nucleation the perturbation to cirrus cloud properties is so high that it can be only matched by nucleation events. Even if contrail formation is happening close to the temperature threshold, contrail formation can significantly alter cirrus properties.

Close to the contrail formation threshold the number of ice crystals increases steeply with increasing distance from the threshold conditions (Fig. 1). This means that even though changes in the temperature threshold for contrail formation are moderate (Fig. 4), amounting to a couple of degrees maximally, they can have a significant impact on contrail ice nucleation when the ambient temperature is close to the temperature threshold for contrail formation and when the cirrus IWC is large. Above 11 km, ambient temperatures are always more than 5K below the contrail formation threshold (Fig. 4a, b) and nucleation rates are high so that a change in the formation threshold would have relatively little impact on nucleation rates. At typical cruise levels between 10.3 km and 10.8 km, the ambient temperature lies often well below and occasionally close to the contrail formation threshold. Due to the smaller difference between ambient temperatures and the contrail formation threshold temperature, fewer ice crystals nucleate (Fig. 1; Fig. 5e, h; Fig. 6e, h). If all emitted soot particles would form an ice crystal then the grid box mean ice crystal number concentration would reach approximately $1.5*10^8$ m$^{-3}$. At typical cruise levels contrail ice nucleation leads commonly to grid box mean ice crystal concentrations of 1.2 x$10^8$ and 1.3 x$10^8$ m$^{-3}$on the 24th (Fig. 5b) and 26th of April 2013 (Fig. 5a), respectively, but on the 24th significantly lower nucleation rates are also fairly typical. The probability of changes in the contrail ice crystal concentration, $\Delta n_i$, is largest for values of $\Delta n_i$ around $10^4$ m$^{-3}$ but changes can reach on the 26th values of close to $10^7$ m$^{-3}$, that is close to 10% of the contrail ice crystal concentration when disregarding the impact of cirrus ice crystals. Changes in the ice nucleation can also be negative due the effect of sublimation and deposition on cirrus ice crystals mixed into the aircraft plume. We estimate that in about 22% and 14% of cloudy grid boxes nucleation rates are reduced on the 26th and 24th, respectively. Contrail ice number concentration are reduced by up to $10^4$ m$^{-3}$, several orders of magnitude smaller than the maximum possible increases in contrail ice number concentration. We will therefore discuss in the following mainly the increases in ice nucleation due to the presence of cirrus ice crystals.

At around 9.7 km height the ambient temperature lies mostly within 5K of the contrail formation threshold (Fig. 4 a, b) and contrail ice nucleation leads to ice number concentrations, that lie typically between 4.0x$10^7$ and 1.1x$10^8$ m$^{-3}$ on the 26th April (Fig. 6a) and between close to 0 and 1.3 x$10^8$ m$^{-3}$ on the 24th April (Fig. 6b). Absolute and relative changes in contrail ice nucleation due to the impact of cirrus ice crystals sublimated in the combustor or mixed into the plume are significantly larger

than at higher flight levels (Fig. 6a,b) because the atmosphere is generally closer to the contrail formation threshold and the
cirrus IWC is on average higher (Fig. 5c, e and 6c, e and 2a, d). The probability for changes in the ice number concentrations
is highest for values ranging between $10^4$ and $10^6$ m$^{-3}$ but changes larger than $10^7$ m$^{-3}$ can also occur on the 26$^{th}$, that is
significantly larger than on the main cruise level. Negative changes in contrail ice number concentrations due the impact of
cirrus ice crystals are less common than on the main flight level and can be found in about 7% of the grid boxes on both the
26$^{th}$ and 24$^{th}$. This means that the change in contrail ice nucleation due to cirrus ice crystals can induce a change in the ice
nucleation of up to the same order of magnitude as the contrail ice nucleation when neglecting the impact of cirrus ice crystals.
On the 24$^{th}$ April the change due to the impact of cirrus ice crystals remains significantly lower in line with the lower ice water
content on that day.

On both the 26$^{th}$ and 24$^{th}$ of April 2013 large changes in the contrail ice number concentration are connected with large cirrus
ice water content (Fig. 5c and 6c), a high number concentration of cirrus ice crystals (Fig. 5d and 6d) and with low ice saturation
ratios (Fig. 5f and 6f). Large scale lifting appears to lead to the freezing of water droplets and/or to homogeneous nucleation
events. The resulting large ice crystal number concentrations lead to an efficient relaxation of ice supersaturation to saturation
values. In areas of lower ice crystal number concentrations, ice supersaturation can be larger (Fig. 5 d, f and 6 d, f). The larger
ice saturation ratio in those areas leads to high contrail ice nucleation rates and low corrections of this nucleation rate. On the
24th April 2013, contrail ice nucleation is lower than on the 26$^{th}$ due to the higher temperatures (Fig. 5h) and the change in ice
nucleation is lower due to the cirrus clouds containing less ice water (Fig. 5g and figure 2) and fewer ice crystals.

Absolute changes in contrail ice nucleation may often be relatively small when compared to ice number concentrations within
contrails that form further away from the formation threshold but they are still high when compared to naturally formed cirrus
clouds. Therefore, contrail ice nucleation and the changes introduced by the sublimation of cirrus ice crystals and the deposition
on them have a significant impact on cirrus properties.

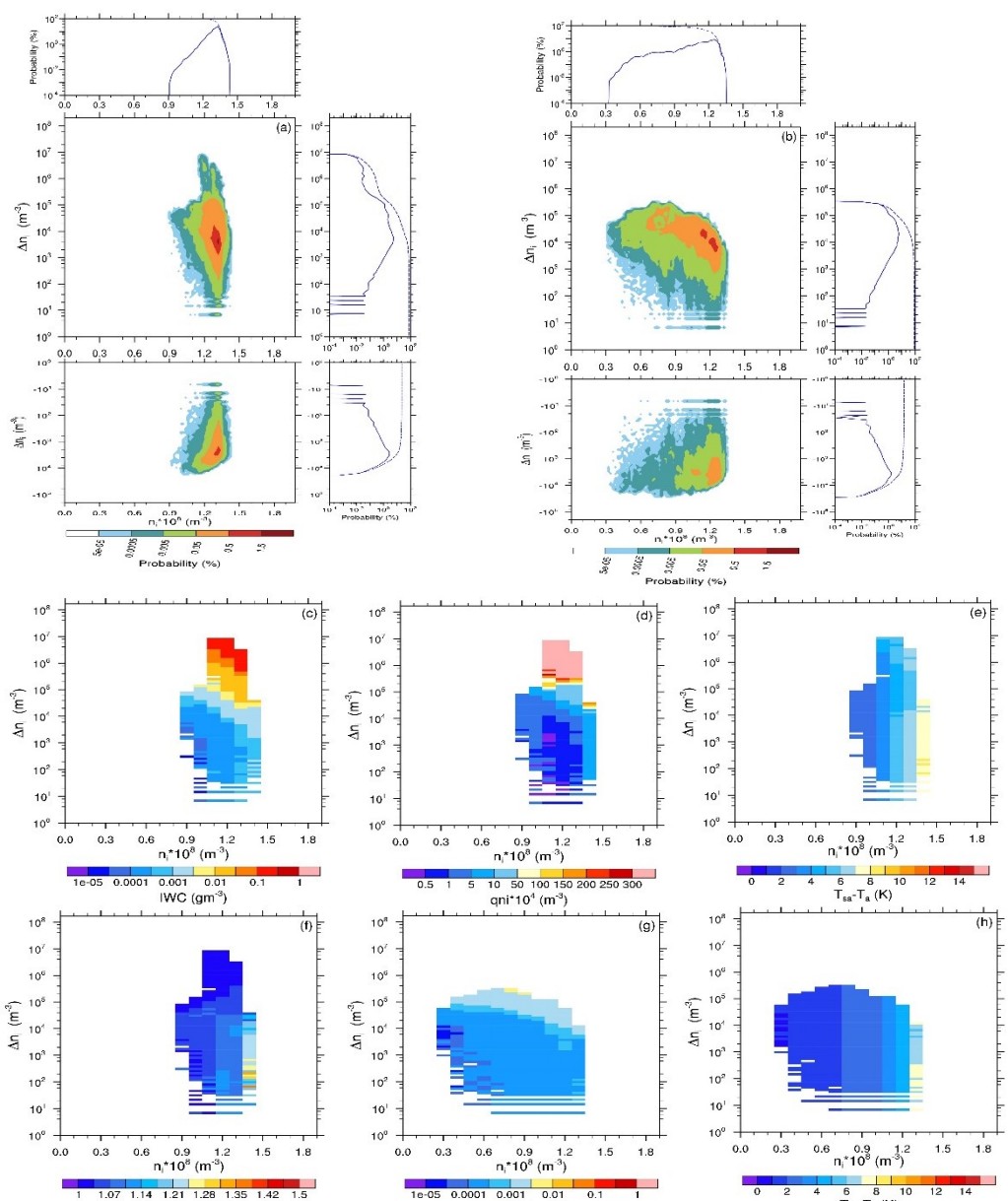

**Figure 5: Joint probability distribution of grid mean ice crystal number concentration due to contrail ice nucleation, $n_i$, and its change due to the sublimation of and deposition on cirrus ice crystals, $\Delta n_i$, for current soot number emissions, $2.5*10^{15}$ kg-fuel$^{-1}$, and for altitudes from 10.3km to 10.8km (250 hPa to 225 hPa) on the (a) 26$^{th}$ April and (b) 24$^{th}$ April 2013. Additionally, the PDF of ice nucleation (solid) and the associated cumulative PDF (dashed) when neglecting the impact of natural cirrus ice crystals (top) and its change due to the sublimation of cirrus ice crystals (right) is shown. Mean cirrus cloud properties for the combination of $n_i$ and $\Delta n_i$ (c,g) IWC, (d) qni, (e, g) difference between temperature formation threshold and ambient temperature and (f) ice saturation ratio. (c), (d), (e) and (f) for cirrus cloud properties on the 26$^{th}$ April 2013 6am and (g) and (h) for the 24$^{th}$ April 2013 5pm. Mean ice cloud properties are shown only for the more common positive changes in ice nucleation. If all emitted soot particles would form an ice crystal then the ice crystal number concentration within the grid box, $n_i$, would reach approximately $1.5*10^8$ m$^{-3}$.**

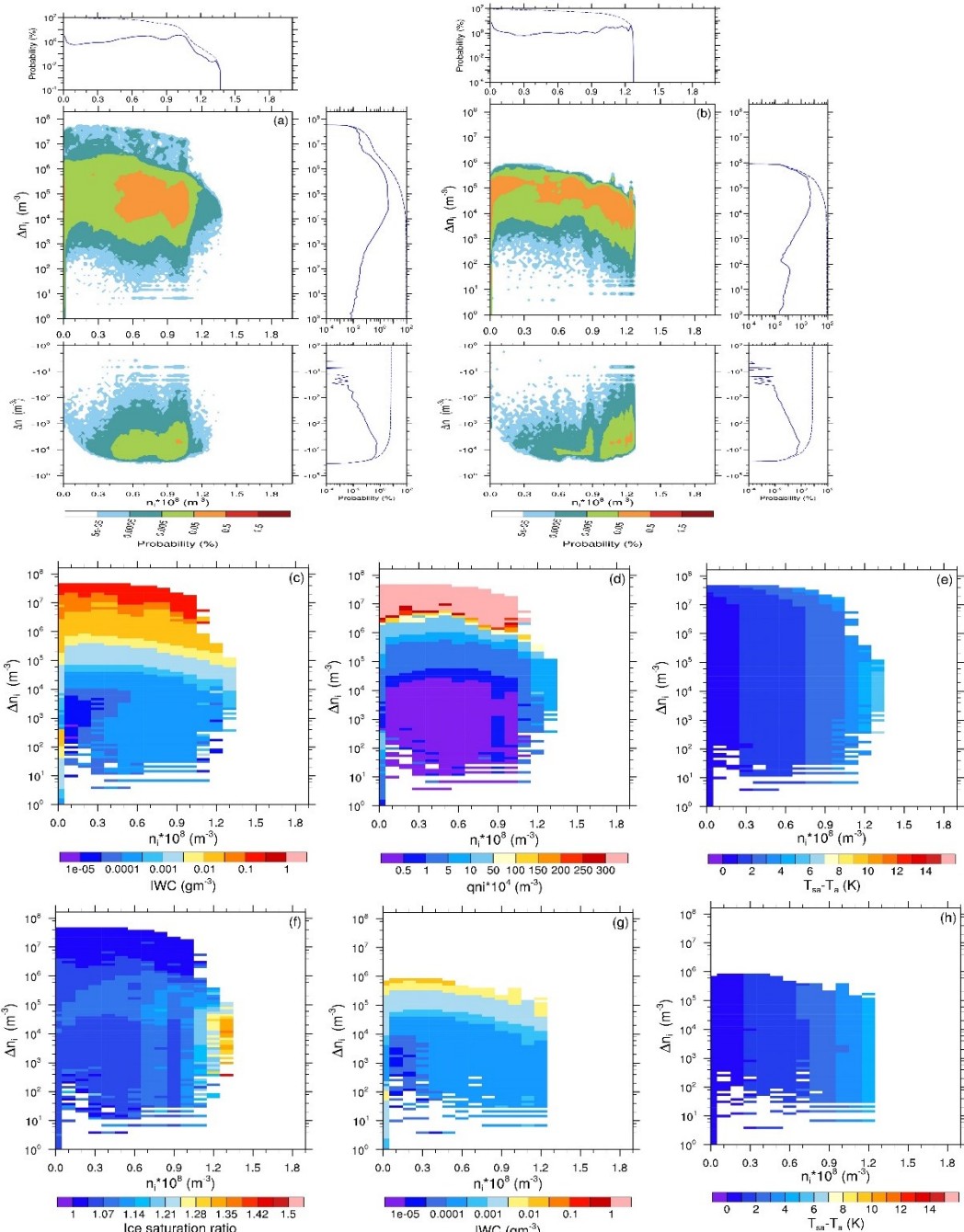

**Figure 6: As figure 5 but for altitudes ranging from 9.6km (280 hPa) to 9.8 km (270 hPa).**

### 3.3 Impact of the pre-existing cirrus on ice crystal loss during the vortex phase

Here we analyze the contrail ice crystal loss during the vortex phase and the impact of pre-existing cirrus ice crystals, that get mixed into the plume before vortex descent, on the ice crystal loss. Generally, we expect low survival fraction since the saturation ratio within cirrus is often close to 1, in particular if cirrus ice crystal concentrations are high. As described in section 2.2.4, cirrus ice crystals have an impact on contrail ice crystal survival due to the competition of cirrus and contrail ice crystals for water vapor deposition. The lower the water vapor mass that is deposited on contrail ice crystals the faster the contrail ice crystals may sublimate completely during vortex descent. Furthermore, the sublimation of cirrus ice crystals can increase the survival fraction of contrail ice crystals during vortex descent because cirrus ice crystals sublimate together with the contrail ice crystals in the ice sub-saturated descending vortices, thereby increasing the relative humidity. This increase in relative humidity within the vortices can lead to a decrease in the ice crystal loss and therefore an increase in the ice crystal survival fraction.

We have analyzed the impact of cirrus ice crystals on the survival of contrail ice crystals, connected with diffusional growth shortly after nucleation and with sublimation within the descending vortices, during the contrail's vortex phase separately for contrails that form more than 5K below the temperature threshold for contrail formation (Fig. 7) and for contrails that form closer to the formation threshold (Fig. 8). We calculate the survival fraction depending on the number of nucleated ice crystals which is itself dependent on the presence of cirrus ice crystals. We test the sensitivity of our results, varying the soot number emission index and the static stability, assuming an atmospheric stability of $0.012 s^{-1}$ a value that is slightly higher than the average in the upper troposphere.

### 3.3.1 Far-below-threshold case

On the 24th of April 2013, it is mainly the atmospheric levels above 11 km and a few hundred meters below where the temperature is more than 5K below the contrail formation threshold (Fig. 4b). At those levels the fraction of ice crystals surviving the vortex phase, when neglecting the impact of natural cirrus ice crystals, is very low and its change caused by the presence of natural cirrus ice crystals is practically zero (not shown). Only in nearly 1% of grid boxes more than 20% of ice crystals survive (Fig. 7a) due to the low ambient ice supersaturation

On the 26[th] April, the temperature on the atmospheric levels above about 10 km is more than 5K below the contrail formation threshold (Fig. 4a). At those levels ice supersaturation and cirrus ice water content (Fig. 2) are significantly larger than on the 24[th] April due to the large-scale rising motion connected with the frontal system. Therefore, ice crystal survival fractions and their change due to the impact of cirrus ice crystals are larger (Fig. 7b). When sampling all grid boxes that have temperatures more than 5K below the contrail formation threshold, the probability of low survival fractions of up to 10% is highest but survival fractions of nearly 75% can be also found. In about 1% of grid boxes survival fractions exceed 40%. The impact of cirrus ice crystals on the survival fractions is low, surpassing only very seldomly values of only 0.05%. Negative changes in

the survival fraction are even smaller. In the following we study the dependency of the survival fraction and its change due to
the impact of natural cirrus ice crystals on the various parameters discussed in Sect. 2.

**Sensitivity to soot number emissions and static stability**

A reduction in soot number emissions leads to fewer ice crystals nucleating within the aircraft plume (Kärcher et al., 2015). A
smaller number of contrail ice crystals within a plume with unchanged total water content leads to larger contrail (and cirrus)
ice crystals and to a larger fraction of contrail ice crystals surviving the vortex descent (Unterstrasser, 2016). This explains the
increase in the probability of high survival fractions and in the maximum survival fraction for the decrease in soot number
emissions by 50% (Fig. 7c) and by 80% (Fig. 7d). In 1% of grid boxes survival fractions exceed 65% and in 10% of grid boxes
survival fractions exceed 30% at 80% reduced soot number emissions. Furthermore, the decrease in the soot number emissions
affects the change in the survival fraction due to the natural cirrus ice crystals. The change in the ice crystal survival fraction
increases for decreasing soot number emissions but stays nearly always below 0.5% even for 80% reduced soot number
emissions. Only for survival fractions of close to zero, is the change in survival fraction due to the presence of cirrus ice
crystals larger, reaching values of more than 1% in approximately 0.08% of grid boxes. On the 24th of April 2013, the change
in the survival fraction caused by the presence of natural cirrus ice crystals is close to zero even for 80% reduced soot emissions
(Fig. 7a) due to the low IWC within the very thin cirrus (Fig. 2). The change in the survival fraction due to cirrus ice crystals
can be negative or positive with positive changes slightly larger. Since the change in survival fraction due to cirrus on 24th
April 2013 is almost negligible we do not show any further analysis for this day.

The atmospheric stability determines the maximum vertical displacement of the wake vortices with high stability limiting the
descent of the vortex. A low stability leads to a large descent of the vortex and therefore to a large decrease in relative humidity
within the vortex and a decreased ice crystal survival fraction. We have analyzed the effect of atmospheric stability on the
survival fraction of contrail ice crystals by lowering the upper tropospheric Brunt-Väisälä frequency to a value of $0.005s^{-1}$ (Fig.
7c). A larger fraction of the contrail ice crystals sublimate during vortex descent in a weakly stable atmosphere (Unterstrasser,
2016, Fig. 7c). The probability of high survival fractions of contrail ice crystals decreases and the probability of large changes
in the survival fraction due to the presence of natural cirrus ice crystals increases very slightly. The probability of large changes
in the survival fraction at survival fractions of close to zero is significantly reduced in a weakly stable atmosphere.

Overall the impact of cirrus ice crystals on the survival of contrail ice crystals in the vortex phase is extremely limited. Only
at survival fractions close to zero the impact can reach values of 1%. At those low survival fractions the uncertainty in the
survival fraction is largest.


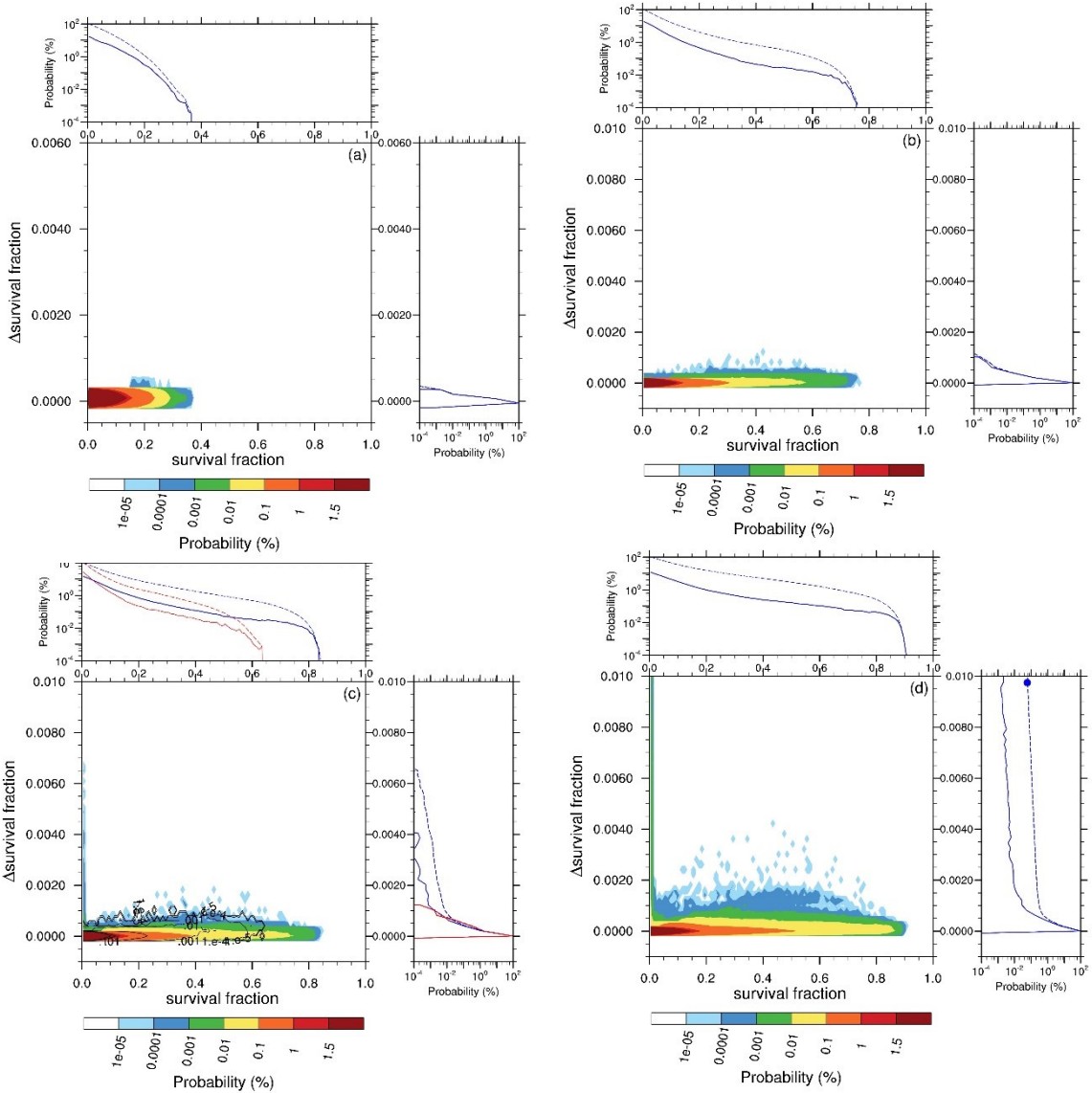

**Figure 7: Joint probability distribution of contrail ice crystal survival fraction during the vortex phase when neglecting the impact of cirrus ice crystals and its change due to the presence of cirrus ice crystals for (a) the 24th April for 80% reduced soot number emissions and for the 26th April 2013 for (b) current soot number emissions, 2.5\*10^15 kg-fuel^-1, (c) 50% reduced soot number emissions and (d) 80% reduced soot number emissions for cases when ambient temperature is at least 5K below the formation threshold temperature. Additionally, the PDF of the fraction of surviving ice crystals (solid) and the associated cumulative PDF (dashed) when neglecting the impact of natural cirrus ice crystals (top) and its change due to the presence of cirrus ice crystals (right) is shown. A Brunt-Väisälä frequency of 0.012 s^-1 (strong stability) has been assumed for all cases. In (c) red lines indicate the probabilities when reducing the Brunt-Väisälä frequency to 0.005 s^-1 (weak stability). The probability isolines in the weak stability case vary between 1x10^-5 to 1.5 % as in the colour bar. Note that in (a) the scale of the y-axis is changed compared to the other figures. The dot in the probability distribution of the change in survival fraction (d, right) indicates the probability of changes in the survival fraction that are larger than 1%.**

**Impact of cirrus cloud properties**

The strength of the impact of natural cirrus ice crystals on the fraction of contrail ice crystals surviving the vortex phase
depends not only on the number of contrail ice crystals forming and on the atmospheric static stability but also on the cirrus properties, in particular the cirrus ice crystal number concentration and ice crystal sizes (equation 5). Fig. 8 shows the cirrus properties, IWC, ice crystal number concentration and sizes and in-cloud ice supersaturation, for each set of survival fraction of contrail ice crystals and its change due to the presence of natural cirrus ice crystals for the 26[th] April. A soot number emission of $0.5*10^{15}$ kg-fuel$^{-1}$ and a Brunt Väisälä frequency of $0.012s^{-1}$ have been prescribed. As expected (see Sect. 2.2.3), the in-
cloud saturation ratio with respect to ice affects the survival fraction of the contrail ice crystals with high ice supersaturation leading to high contrail ice crystal survival fractions (Fig. 8c). If ambient air is close to saturated with respect to ice (saturation ratio close to 1.0) then the contrail ice crystals have only a low ice water mass and adiabatic warming in the descending vortices will lead to low survival fractions and only the ice crystals within the secondary vortex can survive. The pre-existing cirrus has the largest impact on the survival of contrail ice crystals within the vortex phase if the cirrus ice water content and the ice
crystal number concentrations are high and resulting ice crystal sizes are small (Sect. 2.2.4; Fig. 8 a,b,d). When cirrus ice crystals are small then the difference between the size of cirrus and contrail ice crystals is small leading to smaller differences in the diffusional growth of the ice crystals. Therefore, contrail ice crystals grow larger so that the sublimation of cirrus ice crystals within the descending vortices can have a larger impact on contrail ice crystal survival. Furthermore, the large number of cirrus ice crystals that is connected with a large ice water content can effectively moisten the descending vortex and lead to
reduced contrail ice loss. Large cirrus ice crystal number concentrations are commonly connected with relative humidity around ice saturation. Therefore, areas with high cirrus ice crystal number concentrations are often areas where the contrail ice crystal survival fraction (when neglecting the impact of cirrus ice crystals) is low (Fig. 8b) and when they are connected with relatively high IWC (Fig. 8a, b), the pre-existing cirrus has a comparatively large impact on the contrail ice crystal survival fraction. Survival fractions of more than 50%, when neglecting the impact of cirrus ice, are usually connected with ice
saturation ratios above around 1.25. In those areas high cirrus ice crystal numbers are relatively uncommon and hint at recent ice nucleation events presumably connected with the fast lifting of moist environmental air within the frontal system. In areas with survival fraction around 50%, high number concentrations of small cirrus ice crystals and a large IWC lead only occasionally to changes in the survival fraction of more than 0.2%-0.4%. Negative changes in the survival fraction are always connected with few large ice crystals.



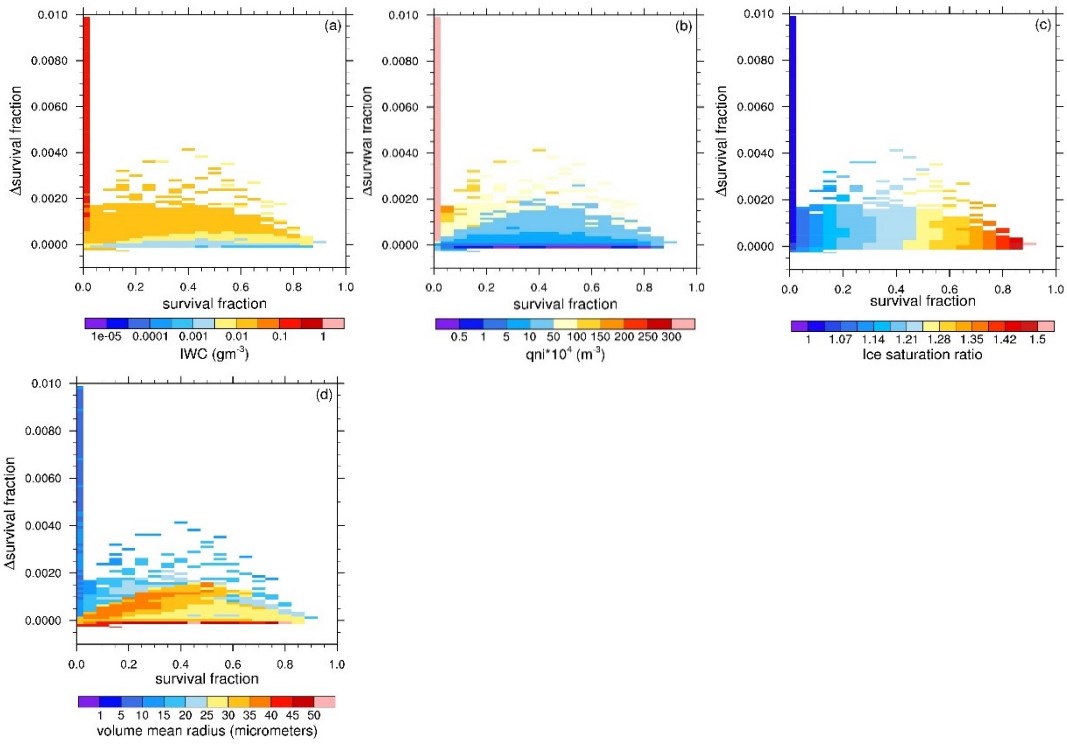

**Figure 8: Ice cloud properties for combinations of contrail ice crystal survival fraction during the vortex phase when neglecting the impact of cirrus ice crystals and its change due to the impact of cirrus ice crystals, Δsurvival fraction, for the 26th April for cases of contrail formation more than 5K below the contrail formation threshold. Color coded are the cloud properties of the pre-existing cirrus, IWC (a), cirrus ice crystal number concentration (b), in-cloud ice saturation ratio (c) and cirrus ice crystal radius (d). A Brunt-Väisälä frequency of 0.012s$^{-1}$ and soot number emissions of 0.5*10$^{15}$ kg-fuel$^{-1}$ are assumed.**

### 3.3.2 Close-to-threshold case

When contrails form close to the contrail formation threshold, large survival fractions (when neglecting the impact of cirrus ice crystals) of contrail ice crystals are much more common than when contrails form at temperatures far below the contrail

formation threshold (Fig. 7d, 9). When temperatures are close to the formation threshold, in about 10% of the grid boxes survival fractions are larger than 70% whereas in the far-from-threshold cases the same percentage of grid boxes exceeds survival fractions of only 30%.

The changes in the survival fraction, due to the impact of cirrus ice crystals, can be significantly larger than in the close to formation threshold cases. In 0.01% of cases the change in the survival fraction exceeds 2-3% but changes can also reach

occasionally values of 5-10% but connected probabilities are extremely small. The probability of a reduction in the survival

fraction of contrail ice crystals due to the presence of cirrus ice crystals is larger than in the far from threshold cases but the changes themselves remain small compared to the increases in the survival fraction.

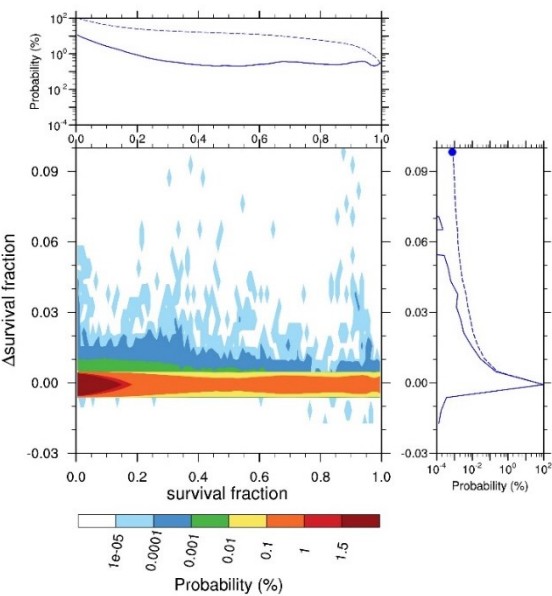

**Figure 9: Joint probability distribution of contrail ice crystal survival fraction during the vortex phase when neglecting the impact of cirrus ice crystals and its change due to the impact of cirrus ice crystals for the 26th April 2013 when contrails form closer than 5K from the contrail formation threshold. Additionally, the PDF of the fraction of surviving ice crystals (solid) and the associated cumulative PDF (dashed) when neglecting the impact of natural cirrus ice crystals (top) and its change due to the presence of cirrus ice crystals (right) is shown. A Brunt-Väisälä frequency of 0.012 s$^{-1}$ (strong stability) and soot number emissions of $0.5*10^{15}$ kg-fuel$^{-1}$ have been prescribed. The dot in the cumulative probability distribution of the change in survival fraction (right) indicates the probability of a change in the survival fraction that is larger than 10%.**

Studying the reasons for the change in the survival fraction of contrail ice crystals due to cirrus ice crystals when contrails form close to the contrail formation threshold is complicated by the fact that the survival is additionally dependent on the

number of nucleated contrail ice crystals which varies strongly depending on the atmospheric state. Assuming fixed soot number emissions, the ambient atmospheric state controls contrail ice nucleation while the survival of the ice crystals during the vortex phase is dependent on atmospheric variables and on the contrail ice nucleation.

High survival fractions are found in areas where the in-cloud ice saturation ratio is high (figure 10c) and/or where contrail formation occurs close to the formation threshold so that the number of nucleated contrail ice crystals is very low (figure 10d).

Due to the low number of contrail ice crystals, the ice crystals are larger and the survival fraction within the vortex phase is increased, similar to the increased survival fractions when reducing soot number emissions. The fact that survival fractions can be high either when the ice supersaturation ratio is high or when it is average but very few contrail ice crystals were formed obscures the strong influence that ice supersaturation has on the survival of ice crystals (Fig. 10c).

The change in the survival fraction is primarily a function of how close to the formation threshold contrails were formed and,

therefore, of the $AEI_i$ (Fig. 10d). The closer contrail formation happens to the formation threshold the fewer ice crystals are formed, so that the impact of the pre-existing cirrus on the survival fraction can be relatively large. Changes in the survival

fraction of contrail ice crystals are practically zero for high $AEI_i$ ($> 10^{13}$ kg-fuel$^{-1}$). At high survival fractions (when neglecting the impact of cirrus ice crystals) changes in the survival fraction can be large only if the cirrus consists of very few ice crystals. When contrail ice survival fractions are close to zero, cirrus with many small ice crystals can lead to large changes in the contrail ice crystal survival fraction.

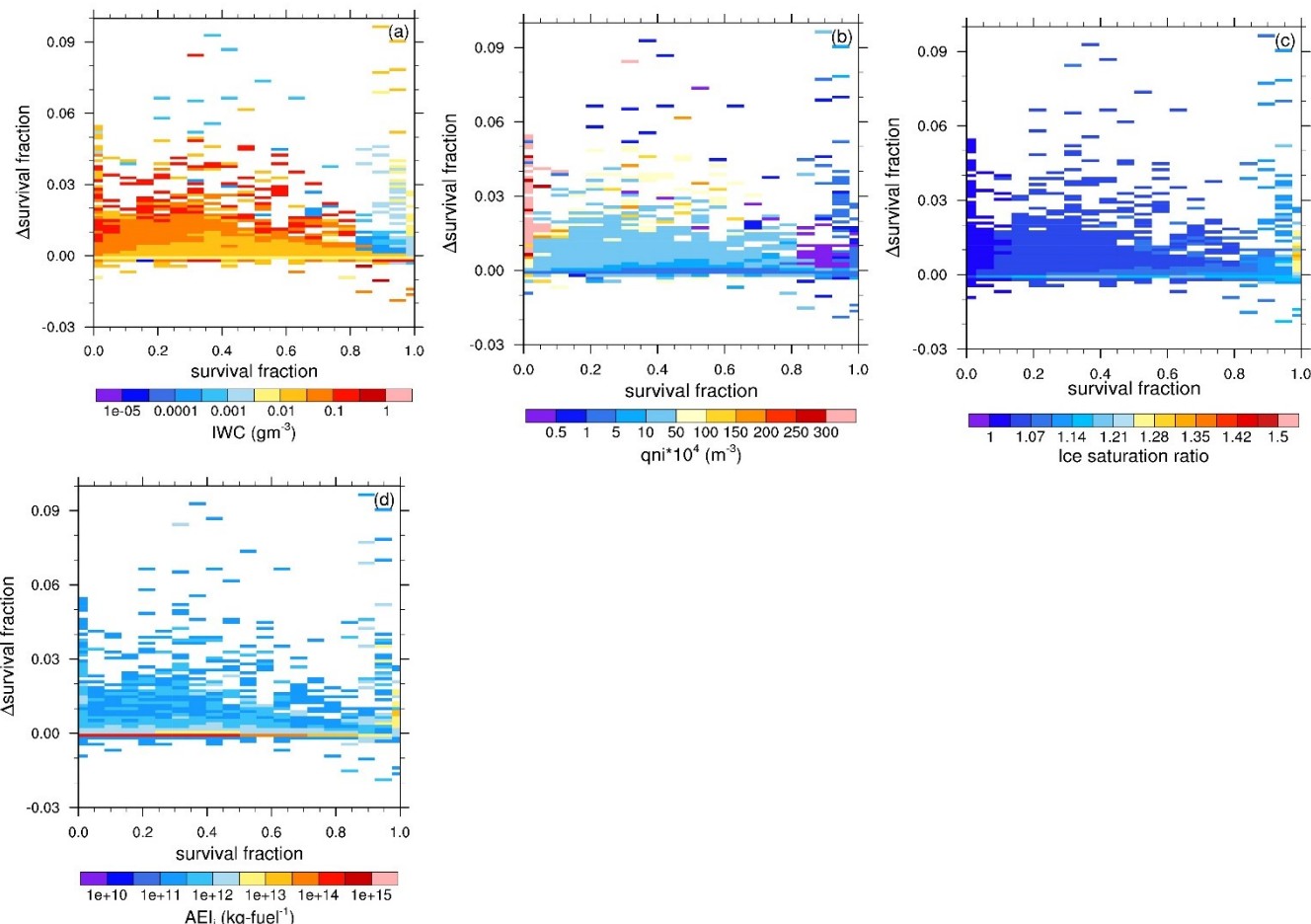

**Figure 10: Ice cloud properties for combinations of contrail ice crystal survival fraction during the vortex phase when neglecting the impact of cirrus ice crystals and its change due to the impact of cirrus ice crystals, Δsurvival fraction, for the 26th April for contrail formation at temperatures closer than 5K from the contrail formation threshold. Color coded are the cloud properties of the pre-existing cirrus, IWC (a), cirrus ice crystal number concentration (b), in-cloud ice saturation ratio (c) and 'new' apparent emission index of contrail ice crystals (d) with 'new' indicating that changes in the nucleation due to the presence of cirrus ice crystals have been considered. A Brunt-Väisälä frequency of 0.012s$^{-1}$ and soot number emissions of 0.5\*10$^{15}$ kg-fuel$^{-1}$ are assumed. AEI$_i$ of 5\*10$^{14}$ translates into qn$_i$=3\*10$^7$m$^{-3}$.**

## 4 Impact of contrail formation on ice crystal number concentrations

Contrail formation within a cirrus cloud leads to changes in the microphysical properties which then have implications for the further development of the cirrus, for its life time and optical properties and, therefore, its impact on climate. In figure 11 we show the change in ice crystal numbers of the cirrus cloud field on the 26$^{th}$ April 2013 at around 10.5km and 9.7km in cases when contrail formation conditions were met, which is in both altitude ranges in about 60% of around 10$^6$ grid boxes in which contrail formation conditions are met. We estimate that in both altitude ranges at around 10.5km and 9.7km, in approximately 90% and 83% of cases the impact on the cirrus was to increase ice crystal number concentrations, respectively. Changes in ice crystal number concentrations approximately 5 minutes after emission are most likely to range between 10$^6$m$^{-3}$ and 10$^7$m$^{-3}$ irrespective of the cirrus ice crystal number concentration (figure 11). In lower levels changes are slightly lower than in upper levels. Changes in ice crystal number concentrations display a small shift towards larger changes for higher initial cirrus ice crystal number concentrations which is likely caused by the impact of the pre-existing ice crystals on contrail ice nucleation discussed in section 3. Cirrus ice crystal number concentrations range between 10$^2$m$^{-3}$ and 10$^5$m$^{-3}$ before air traffic and the formation of contrails within cirrus is most likely to change ice crystal number concentrations by 2 - 4 orders of magnitude. We do not estimate negative changes in ice crystal number concentrations mainly for two reasons. First, we do not simulate the impact of air traffic on cirrus properties when no contrails can form and, second, we find that negative changes in ice crystal number concentrations, caused by contrails forming but nearly all ice crystals sublimating within the vortex phase, are connected with a high uncertainty that may be of a similar order of magnitude as our estimates, which indicate decreases of maximally 10$^3$m$^{-3}$ at a cirrus ice crystal number concentration of 10$^4$m$^{-3}$.

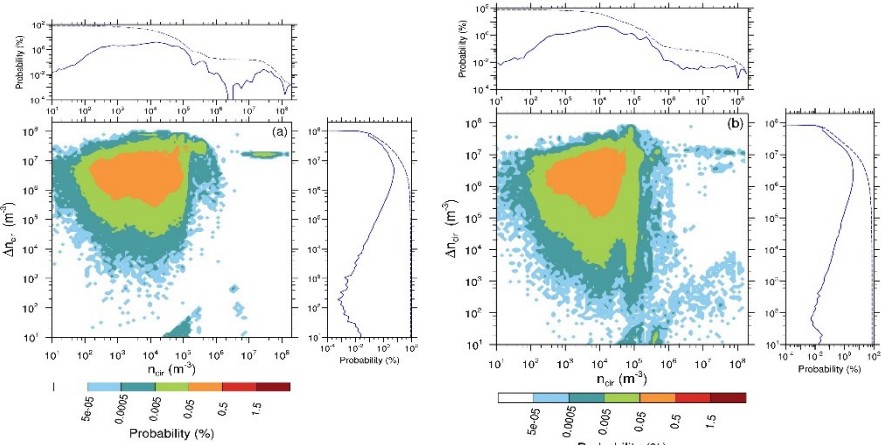

**Figure 11: Joint probability of cirrus ice crystal number concentrations, n$_{cir}$, and the aviation induced change in ice crystal numbers Δ n$_{cir}$, for the cirrus cloud field on the 26$^{th}$ April 2013 for current soot number emissions, 2.5*10$^{15}$ kg-fuel$^{-1}$, and (a) for altitudes from 10.3km to 10.8km (~250 hPa to ~225 hPa) and (b) for 9.6km to 9.8km (~280 hPa to 270 hPa). Only 60% of cloudy grid boxes at the two altitude ranges are analysed in which the contrail formation conditions are met. Grid mean concentrations and their changes are displayed. Cumulative probabilities of Δ n$_{cir}$ reach values below 100% because we do not show negative changes here as they are lower and uncertainty in those simulated changes is high.**

## 5 Summary and comparison with literature

The impact of pre-existing cirrus on contrail formation, the change in the contrail formation criterion and in ice crystals survival in the vortex phase, has been discussed before by Gierens (2012). In agreement with Gierens (2012) we find that the sublimation of the ice crystals that get sucked into the aircraft's engine increase plume relative humidity only slightly. The impact of ice crystal sublimation within the engine on the aviation induced change in humidity in the aircraft plume amounts to maximally a few percent in case of medium to thick cirrus and to less than 1% in the case of the very thin cirrus. This agrees with the 'order of percent' estimate of Gierens (2012). When additionally considering the impact of the cirrus ice crystals that are mixed into the plume after emission, the sublimation, as long as the plume is ice subsaturated, and deposition when supersaturated increases the change in the aviation induced water vapor content on average. Most of the time sublimation exceeds deposition on mixed-in cirrus ice crystals. Changes in the aviation induced water vapor content of the plume due to the impact of cirrus ice crystals one second after emission can amount to 10%-20%. Our estimate of the impact of cirrus on the aviation induced changes in water vapor, including the impact of mixed-in cirrus ice crystals, are higher than the estimate of Gierens (2012) since he did not consider the additional impact of mixed-in cirrus ice crystals.

Consistent with us estimating a larger plume moistening effect due to the pre-existing cirrus ice crystals, our estimate of the impact on the contrail formation threshold is also larger than in Gierens (2012). We estimate that the increase in the plume relative humidity leads to changes in the contrail formation threshold of up to 1.5 - 2 K as opposed to Gierens estimating that cirrus ice crystals have no significant impact. Nevertheless, we agree that in large parts of the cirrus cloud field the presence of cirrus ice crystals does not impact the contrail formation criterion and contrail ice nucleation significantly.

Ice nucleation is estimated to be significantly changed due to the impact of cirrus ice crystals in particular when the cirrus IWC and ice crystal number concentration is large. Changes can be positive (increasing contrail ice nucleation) or negative (decreasing contrail ice nucleation) with positive changes significantly larger and more common. While for far-from-threshold cases changes can amount to nearly 10% of the maximum contrail ice crystal number concentration, at temperatures close to the formation threshold, contrail ice nucleation can be often lower and changes in contrail ice nucleation due to the impact of cirrus ice crystals larger. This means that changes in contrail ice nucleation due to pre-existing cirrus are larger in lower flight levels and in the tropics because in those areas the IWC is usually larger and contrail ice nucleation lower (Bier and Burkhardt, 2019). We conclude that the sublimation of cirrus ice crystals in the engine and the impact of cirrus ice crystals mixed into the plume can have a significant impact on contrail formation. Nevertheless, we agree with Gierens (2012) that the use of alternative fuels and the associated increase in water vapor emissions, and in particular the associated decrease in soot number emissions, have significant implications for ice nucleation which is why they are regarded to be an important mitigation method (Burkhardt et al., 2018).

When analysing the impact of cirrus ice crystals on the survival of contrail ice crystals brought about by changes to water vapor deposition on contrail ice crystals and to sublimation of contrail ice crystals in the descending vortices, we find that, for current day soot emissions, changes in the survival fraction are of the order of a few tenth of a percent when contrails form more than 5K below the formation threshold in a stably stratified atmosphere. Changes in the survival fraction can be both positive or negative with positive changes slightly larger. They are largest in cirrus that comprise large number concentrations of small cirrus ice crystals and a large ice water content or in cases of low contrail ice nucleation. When reductions in soot number emissions are introduced, e.g. caused by the introduction of alternative fuels, the larger contrail ice crystal sizes lead to a larger change of the survival fraction due to the presence of cirrus ice crystals because the difference between the contrail and cirrus ice crystal sizes gets smaller. Even for 80% reduced soot number emissions changes in the survival fraction of ice crystals are low reaching maximally 1% when otherwise contrail ice crystal survival would have been close to zero. When contrail formation happens close to the formation threshold, the change in the ice crystal loss can be larger than in far-from-threshold cases, but the probability of those large changes is very low and furthermore the uncertainty in those estimates is high. Absolute changes in the survival fraction amount maximally to about 10% when assuming 80% reduced soot number emissions and a stably stratified atmosphere and are connected with low $AEI_i$, or with cirrus consisting of many small ice crystals. Despite us using a very different measure for estimating the impact of cirrus ice crystals on the contrail ice crystal survival fraction than Gierens (2012), we agree that the impact is mostly extremely low. Changes in the survival fraction of contrail ice crystals due to the impact of cirrus ice crystals larger than a few percent are extremely uncommon and are mostly found in lower flight levels. Contrail formation within slightly aged contrail cirrus may lead to larger changes.

Percentages of the cirrus volume in which contrail formation is affected by pre-existing cirrus are very much dependent on the definition of a cirrus cloud. By including extremely thin cirrus in our analysis (using an IWC threshold of $10^{-11}$kg m$^{-3}$) the fraction of the cirrus volume with high IWC and, therefore, with a significant impact of the pre-existing cirrus, is low. Restricting our analysis to cirrus with larger IWC or larger optical depth, e.g. to only those cirrus that would be visible by eye from the ground, would lead to an increase in the likelihood with which pre-existing cirrus can have an impact on contrail formation.

We find that contrail ice nucleation within cirrus introduces a large change in the cirrus microphysical properties. When comparing cirrus ice crystal number concentrations before air traffic and the impact of air traffic approximately 5 minutes after emission, ice crystal number concentrations commonly increase by 2 to 4 orders of magnitude.

**6 Conclusions**

Contrail formation constitutes a significant perturbation to cirrus cloudiness. Until now it is mainly the impact of contrail formation in cloud free air that has been studied. Recently, satellite observations of cirrus perturbations caused by contrail formation within cirrus resulting in an increase of cirrus optical depth (Tesche et al., 2016) have led to increased interest in the topic. Here we present a contrail scheme, consisting of the estimation of contrail formation conditions and parameterizations

for contrail ice nucleation and contrail ice crystal loss in the vortex phase, within ICON-LEM (Zängl et al., 2015). We adapt the scheme in order to study contrail formation within cirrus and study whether contrail formation is modified due to the impact of pre-existing cirrus ice crystals.

We find that contrail ice nucleation within cirrus often lead to increases in local ice crystal number concentrations by 2 to 4
orders of magnitude even if survival rates of contrail ice crystals are often low. Pre-existing ice crystals can impact contrail ice nucleation and, therefore, ice crystal number concentrations significantly with positive and negative changes possible and positive ones larger and more common. The presence of pre-existing cirrus ice crystals leads often to an increase in the contrail formation threshold temperature. Therefore, contrails may form and cause locally relatively high ice crystal number concentrations when, in the absence of the pre-existing ice crystals, no contrails would form so that only the sublimation of
cirrus ice crystals in the engine and in the subsaturated plume would lead to a change in the ice crystal concentrations. In case of cirrus with a large IWC and ice crystal number concentration, such as the cirrus that has properties representative of medium thick in-situ formed cirrus or fast rising liquid-origin cirrus (Krämer et al. 2020), contrail formation, in particular the contrail formation threshold and contrail ice nucleation, can be noticeably modified. Nevertheless, the pre-existing cirrus ice crystals have in large parts of the cloud field a negligible impact on the contrail formation processes.

The synoptic situation in which the impact is large, e.g. the outflow of a warm conveyor belt connected with frontal activity and the associated steady supply of moisture in the lifting zones, is exactly the synoptic situation in which contrails that form in cloud free air have a long-life time and a large climate impact. The change in cirrus ice crystal numbers due to contrail formation may, therefore, have a long-lasting influence on cirrus optical depth, radiative fluxes and cirrus life times.

Many uncertainties remain in our estimates that are mainly caused by the simple two-moment microphysical scheme of ICON-
770 LEM and by the assumptions made in the parameterizations that we are using. Since we do not resolve the size distribution of ice crystals we can only capture the differences in the diffusional growth of contrail or cirrus ice crystals depending on their mean ice crystal sizes. This means that we cannot estimate how many smaller cirrus ice crystals may be lost with the subsaturated plume. We also do not have any information on ice crystal shapes which would be important for estimating the diffusional growth in particular when cirrus ice crystal sizes are large. For small cirrus and contrail ice crystals the assumption
of sphericity may be reasonably good. The parameterization for ice nucleation of Kärcher et al. (2015) assumes that all contrail ice crystals are formed at the same time and, therefore, does not consider successive ice nucleation which would have an impact on the temporal development of the plume relative humidity. Successive nucleation may result from varying properties of ambient aerosols and from inhomogeneities of the plume. Estimates of soot number emission are highly variable. Furthermore, contrail formation close to the formation threshold is connected with a larger uncertainty than contrail formation
far away from the threshold since details in the plume development may have a large impact, leading to varying ice crystal numbers resulting from slightly different plume evolutions (Lewellen, 2020). The parameterization of ice crystal survival is connected with a large uncertainty in cases of very low $AEI_i$. We would welcome studies using LES with a complex microphysical scheme that could shed more light onto these uncertainties. Such a model could potentially give valuable insights regarding impact of ice crystal size distributions, aerosol concentrations and properties and shape of ice crystals.

We have shown that the large perturbations induced by contrail formation within natural cirrus lead to large modifications of cirrus microphysical properties. It would be important study the life cycle of these disturbances and to estimate the associated changes in optical properties and, consequently, in radiative transfer that need to be included in aviation climate change assessments (Lee et al., 2021). Our study makes a first step in that direction by devising a contrail cirrus scheme with which contrail formation within clouds and their impact on cirrus properties and radiative forcing can be studied in models that do not resolve the plume processes. A complete picture of the climate impact of air traffic including all the climate forcing components together with their uncertainties is also crucially necessary for the evaluation of mitigation options that require calculating the tradeoffs between different climate forcing components. Finally, our work allows to improve the interpretation of cirrus observational data from flight campaigns and remote sensing and adds complexity to discussions about the importance of different ice nucleation pathways for cirrus properties. Contrail induced perturbations in cirrus properties may be important when evaluating models with in-situ measurements or within a data assimilation context.

**Appendix A: Impact of cirrus ice crystals mixed into the plume on the plume water vapor partial pressure before aerosol activation**

The cirrus ice crystals, entrained into the plume during the rapid mixing of plume and ambient air immediately after emission, sublimate in the young, ice subsaturated plume until the plume reaches ice saturation due to the mixing with environmental air. At the beginning of the mixing, the sublimation rate of entrained ice crystals is large since the plume's relative humidity is low and temperature very high. Due to rapid mixing in the plume, the plume's temperature decreases and relative humidity increases. After ice saturation is reached and before aerosol activation, water vapor deposits on the cirrus ice crystals reducing the fast increase in the plume ice supersaturation.

The rate of sublimation / deposition on an ice crystal can be calculated using the diffusional growth equation given in e.g. Pruppacher and Klett (1996) and Lewellen (2012) assuming spherical ice crystals:

$$\frac{dm}{dt} = \frac{4\pi r(S - e^{ak/r})}{L_s^2 K/R_v T^2 + R^v T/e_i(T)D} \tag{6}$$

Where, r is radius of the ice crystal, S is ice supersaturation ratio, D is diffusivity of water vapor, K is thermal conductivity of air and $e_i(T)$ is saturation vapor pressure at temperature T. Sublimation/Deposition depends mainly on the difference between the ambient vapor pressure and the saturation vapor pressure considering the dependence of the saturation vapor pressure on the ice crystal radius (Kelvin effect). In the curvature term we set the Kelvin radius '$a_k$' to $2\times10^{-9}$m, representative for cirrus-type conditions (Lewellen, 2012). The diffusivity of water vapor and thermal conductivity of air is calculated considering that the vapor density in moist air is not continuous right up to the surface of the ice crystal, using equations given in Pruppacher

and Klett, 1996 (equations 13-14,13-20) setting the thermal accommodation coefficient to 0.7 and the mass accommodation coefficient to 0.5 (Kärcher, 2003). We assume spherical particles since we do not have information on the habit of the ice crystals in our model which can lead to an underestimation of the growth of ice crystals. For contrail ice crystals sphericity should be a good assumption while for larger cirrus ice crystals this assumption is often not good. Nevertheless, the assumption of spherical particles in our study may often reasonably good since large changes in contrail ice nucleation and survival are caused by cirrus consisting of many small ice crystals and a large ice water content. That means that at times when we see the largest impacts the assumption of spherical particles is likely a reasonably good one.

Since we calculate contrail formation at a relatively high resolution in a large domain we cannot calculate the temporal evolution of the plume and of the sublimation and deposition on mixed-in cirrus ice crystals within ICON-LEM. Therefore, we estimate the change in the plume properties based on the time integrated sublimation and deposition on cirrus ice crystals. As an example, we calculate for two contrail formation cases, one far from threshold and one close to threshold case, the total ice water mass sublimated from entrained cirrus ice crystals in the subsaturated plume and the water vapor deposited on the cirrus ice crystals once the plume is supersaturated using equation 6 and an equation for plume dilution (Kärcher et al., 2015, equation 12). We choose two sets of ambient conditions and aircraft parameters as in Bier et al. (2021, their table 1; table 1) and specify cirrus properties (table 1) and calculate the temporal evolution of the plume properties and the cirrus ice crystal number concentration and ice water mass. High plume temperatures right after emission cause a large sublimation rate of entrained ice crystals while the number of entrained cirrus ice crystals is low. Close to ice saturation the sublimation rate is low while the number of entrained cirrus ice crystals is large. The resulting time integrated sublimated ice water mass and the ice water mass deposited on the mixed-in cirrus ice crystals is given in table 1.

We approximately estimate the sublimation of the cirrus ice water mass from the plume conditions at $t_{as}=(\frac{2}{3}(t_{sat}-t_0))$ with $t_0$ the time of emission and $t_{sat}$ the time at which ice saturation is reached. For both our example cases, the time integrated sublimated ice water mass from entrained cirrus ice crystals is very close to the estimate when using plume conditions at time $t_{as}$ with deviations lower than 1% (table 1). The deposition of water vapor on cirrus ice crystals in the supersaturated plume has been estimated from the plume atmospheric variables midway between ice saturation ($t_{sat}$) and aerosol activation ($t_{act}$) at $t_{ad}=(\frac{1}{2}(t_{act}-t_{sat}))$. Since we do not have any information from the ICON model on the size distribution of the cirrus ice crystals we cannot estimate the number of ice crystals lost during sublimation within the subsaturated plume. We assume here that 20% of cirrus ice crystals were lost until the plume reached ice saturation. Table 1 shows that the time integrated deposition and the estimates midway between ice saturation and aerosol activation agree reasonably well (table 1).

In order to estimate the range of errors that we make if we estimate the deposition and sublimation in the above way we vary the background relative humidity, cirrus ice crystal numbers and ice water content and calculate in plumes (one far from the contrail formation threshold and one close to it) the time integrated and the approximated sublimation and deposition within the homogeneous plume. We find that errors in estimating sublimation are mostly below 3% and in estimating deposition

around 2%. Larger errors in sublimation and deposition of up to 4-5% we find only for combinations of atmospheric variables that are unlikely to occur such as high relative humidity combined with many cirrus ice crystals and a large ice water content or for many very small ice crystals, with small ice water content and low supersaturation. When estimating contrail formation within ICON we, therefore choose to estimate sublimation of cirrus ice crystals and water vapor deposition on cirrus ice crystals before aerosol activation by approximating them based on the sublimation and deposition rates at $t_{as}$ and $t_{ad}$, respectively. In order to calculate the deposition on cirrus ice crystals before aerosol activation we determine $t_{act}$ by using the contrail ice nucleation parameterization of Kärcher et al. (2015) assuming $t_{act}$ is unchanged by the sublimation and deposition.

**Table 1: Sublimation of cirrus ice crystals and deposition on cirrus ice crystals mixed into a homogeneous aircraft plume after emission and before ice nucleation prescribing engine exit temperature of 580K and the air to fuel ratio $N_0 = 75$ and specifying cirrus ice crystal number concentration of $n_i=2*10^4$ m$^{-3}$ and ice water content of $10^{-5}$ kgm$^{-3}$. The contrail formation threshold is 226.2K. In order to estimate deposition on cirrus ice crystals from the deposition rate at time $t_{ad}$ we assume a reduction in cirrus ice crystal numbers due to the prior sublimation of 20%.**

| Estimating the sublimation and deposition within the plume before contrail formation | | | | |
|---|---|---|---|---|
| **Ambient conditions** | | | | |
| | Far from contrail formation threshold | | Close to contrail formation threshold | |
| Contrail formation threshold temperature | 226.2 K | | 226.2 K | |
| Temperature | 220 K | | 225 K | |
| RH$_i$ | 120% | | 120% | |
| Pressure | 240 hPa | | 240 hPa | |
| **Sublimation of entrained cirrus ice crystals before contrail ice nucleation AEI (kg/kg-fuel) $\left(\frac{dm}{dt}\right)_{sub}$** | | | | |
| Time of reaching ice saturation '$t_{sat}$' | 0.19 s | | 0.271 s | |
| | Temporal evolution | based on sublimation rate at $t_{as}$ | Temporal evolution | based on sublimation rate at $t_{as}$ |
| Total sublimated ice water mass | 8.81135x10$^{-5}$ | 8.88161x10$^{-5}$ (+0.8659%) | 3.71874x10$^{-4}$ | 3.75298x10$^{-4}$ (0.92074%) |
| **Deposition on entrained cirrus ice crystals before contrail ice nucleation (kg/kg-fuel) $\left(\frac{dm}{dt}\right)_{dep}$** | | | | |
| Time at aerosol activation '$t_{act}$' | 0.45 s | | 1.1 s | |
| | Temporal evolution | based on deposition rate at $t_{ad}$ | Temporal evolution | based on deposition rate at $t_{ad}$ |
| Total deposited ice water mass | 8.083842x10$^{-5}$ | 8.28574x10$^{-5}$ (2.49755%) | 3.0446x10$^{-4}$ | 3.099979x10$^{-4}$ (1.8189%) |

## Code Availability

The ICON model is distributed to institutions under an institutional license issued by the DWD. Two copies of the institutional license need to be signed and returned to the DWD.ICON can be then downloaded at https://data.dwd.de. To individuals, the ICON model is distributed under a personal non-commercial research license distributed by the MPI-M (Max Planck Institute for Meteorology). Every person receiving a copy of the ICON framework code accepts the ICON personal non-commercial research license by doing so. Or, as the license states, any use of the ICON software is conditional upon and therefore leads to an implied acceptance of the terms of the Soft-ware License Agreement. To receive an individually licensed copy, please follow the instructions provided at https://code.mpimet.mpg.de/projects/iconpublic/wiki/Instructions_to_obtain_the_ICON_model_code_with_a_personal_non-commercial_research_license.

## Data Availability

Data is used in the figures can be access from the given DOI: (Verma, 2021, https://doi.org/10.5281/zenodo.5730287)

## Author contributions

PV and UB jointly designed the study. PV have performed simulations and analyzed the results. PV and UB jointly discussed scientific results and wrote the manuscript.

## Competing interests

The authors declare that they have no conflict of interest.

## Acknowledgements

We gratefully acknowledge the High Definition Cloud and Precipitation HD(CP)$^2$ project that performed the ICON-LEM simulations on which our study is based. We thank Jan Frederik Engels from DKRZ for technical help implementing the contrail scheme and the air traffic inventory in ICON-LEM. We thank Klaus Gierens, Andreas Schäffler and Simon Unterstrasser from DLR and Axel Seifert and Daniel Reinert from DWD and the whole HD(CP)$^2$ community for valuable discussions. We thank Winfried Beer and Bastian Kern for their technical support at DLR. The paper was significantly improved by the reviews of David Lewellen and an anonymous reviewer and the comment of Bernd Kärcher.

**Financial support**

This work is funded by the research program "High Definition of Clouds and Precipitation for Advancing Climate Prediction" (HD(CP)2) of the BMBF (German Federal Ministry of Education and Research) under grant 01LK1503C. The authors gratefully acknowledge the computing time granted by the German Climate Computing Centre (Deutsches Klimarechenzentrum, DKRZ, Hamburg, Germany).

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
