# Peer review of "Contrail formation within cirrus: ICON-LEM simulations of the impact of cirrus cloud properties on contrail formation"

_Atmospheric Chemistry and Physics, 2021_

## Author Comment (AC4)

**Comments I (italicized), and responses (blue):**

*Any shift in estimates of effective radius forcing of aviation CO2 and non-CO2 emissions has important repercussions for assessing the trade-offs between the long-lived (CO2) and short-lived (contrail cirrus, for that matter) forcing agents [Simpkins, 2020]. This study is an important stepping stone towards research into the global climate effects of in-cirrus contrail formation, a hitherto under-researched aspect of contrail formation.*

*The authors focus on the contrail formation stage and modify an existing contrail parameterization scheme [Kärcher et al., 2015] to account for in-cirrus contrail ice formation by enhancing the water vapor emission index consistent with adding cirrus ice water sublimated within aircraft engines/combustors. Changes in the water vapor emission index as input to the original parameterization should lead to robust results regarding changes of the contrail formation threshold temperature.*

*With regard to the proposed modification, the sublimated cirrus ice water content was added to the mass emission index of water vapor resulting from fuel combustion with an assumed efficiency of 100%, cf. eqs 2+3. It might be appropriate to note that this assumption does not consider vapor losses e.g., on wall surfaces and the effects studied are therefore upper limit estimates.*

*Moreover, this approach does not capture the possible impact of sublimating cirrus ice crystals entrained into the expanding jet plume prior to and alongside contrail formation. The sublimation of entrained cloud ice in the bypass regions away from the jet core might lead to an increase in the plume water vapor mixing that may be larger than that from ice crystals sucked into the engines, based on microphysical studies of contrail ice formation including entrained cirrus ice [Kärcher et al., 1998]. This increase depends on the ambient cirrus ice crystal number-size distribution and temperature. Ideally, a quick yet reasonably accurate estimate of the contribution of a continuously entrained sublimating cirrus ice crystal population to total plume water vapor may be obtained by applying the flow field model described in Kärcher & Fabian [1994].*

*'Warm' contrails are those observed at flight levels at temperatures well above those obtained by the standard thermodynamic model [Jensen et al., 1998], i.e. typically above about 225 K. It is possible that they form in high humidity (possibly cloudy) regions. The formation of such contrails might be explainable by one of the above mechanisms associated with in-cirrus contrail formation, or by both.*

→Thank you for your comment. We have introduced many changes to our study that consider the reviewer's and your comments.

We do now consider the ice crystals that are mixed into the plume after emission. We estimate the sublimation of the cirrus ice crystals and the deposition that is happening after the plume reaches ice saturation and modify the slope of the mixing line accordingly. We added an appendix describing how we estimate the sublimation/deposition on the cirrus ice crystals mixed into the plume based on the diffusional growth equation. Figures and text were modified throughout the paper.

We estimate that in some instances the impact of the preexisting cirrus ice crystals is to reduce contrail ice nucleation while in most occasions ice crystal nucleation is enhanced. The likelihood of this happening depends on the atmospheric conditions.

To our knowledge the H2O content of the air is not changed when going through the engine. Water does not accumulate in the engine.

We do reference now to the 'warm' contrail estimate in Kärcher et al. 1998.

*The authors are aware of the fact that during contrail formation within cirrus, contrail ice crystals get entrained in to jet plumes (line 259f). Consequently, they estimate the sublimation losses that are expected to occur when entrained cirrus ice crystals get captured in downward propagating wake vortices, albeit again with an assumed efficiency of 100%. Equation 5 makes total sense to me, as in the initial phase of sublimation, the total cloud ice crystal number stays constant while the ice water mass is decreasing. However, at some point cloud ice crystal numbers will start to decrease as well, once the smallest ice crystals fully sublimated to their aerosol cores [illustrated in fig S1 in Kärcher & Voigt, 2017]. The point where this happens obviously depends on the mean ice crystal size.*

→We changed our representation of the contrail ice crystal loss in the vortex phase to include the ice crystal growth phase after contrail nucleation and before vortex descent. We treat that phase using the full diffusional growth equation which can lead to significantly reduced contrail ice crystal sizes.

It is true that the microphysical two-moment scheme of ICON LEM (without an explicit representation of the size distribution) is one of the reasons for uncertainty in our simulations. But since the impact of pre-existing cirrus ice crystals on the vortex phase loss is very low, this uncertainty should be of limited importance.

*Due to their smaller mean size, the contrail variables (total number and mean size) will change much faster during sublimation than the corresponding cirrus variables. I was just wondering whether eq 5 accounts for effects of changing number and size of cirrus and contrails during the sublimation process. If not, how much will the application of eq 5 with constant integral radii for contrail and cirrus ice will deviate from estimates where these variables are allowed to change?*

→Our equation 5 is significantly changed now in order to consider the full dependence of diffusional growth on ice crystal sizes. We include now in our estimate of ice crystal loss in the vortex phase the competition of contrail and cirrus ice crystals before vortex descent. We calculate the temporal evolution of both contrail and cirrus ice crystals in order to have a good estimate of ice crystal sizes before the vortex descent. Afterwards we use equation 5 in order to estimate the cirrus ice water that sublimates in the time that it takes the contrail ice crystals to sublimate without updating the ice crystal sizes. Since the changes in the ice crystal survival due to the impact of cirrus ice crystals is very low, we do not think that inclusion of this would have a large effect on the simulations.

References

Simpkins, G.  The climate cost of flying Nat. Rev. Earth Environ., 1, https://doi.org/10.1038/s43017-020-00104-0, 2020

Kärcher, B., Burkhardt, U., Bier, A., Bock, L. & Ford, I.J.  The microphysical pathway to contrail formation J. Geophys. Res., 120, https://doi.org/10.1002/2015JD023491, 2015

Kärcher, B., Busen, R., Petzold, A., Schröder, F.P. & Schumann, U.  Physico-chemistry of aircraft-generated liquid aerosols, soot, and ice particles. 2. Comparison with observations and sensitivity studies J. Geophys. Res., 103, https://doi.org/10.1029/98JD01045, 1998

Jensen, E.J., Toon, O.B., Kinne, S., Sachse, G.W., Anderson, B.E., Chan, K.R., Twohy, C.H., Gandrud, B., Heymsfield, A. & Miake-Lye, R.C. Environmental conditions required for contrail formation and persistence J. Geophys. Res., 103, https://doi.org/10.1029/97JD02808, 1998

Kärcher, B. & Fabian, P. Dynamics of aircraft exhaust plumes in the jet regime Ann. Geophys., 12, https://doi.org/10.1007/s00585-994-0911-9, 1994

Kärcher, B. & Voigt, C. Susceptibility of contrail ice crystal numbers to aircraft soot particle emissions Geophys. Res. Lett., 44, https://doi.org/10.1002/2017GL074949, 2017

---

## Author Comment (AC5)

**Reviewer #1 Comments (italicized), and responses (blue):**

*The general topic of this work -- how formation within natural cirrus may affect contrail properties and thereby their potential impact on climate -- is a potentially important one. The results of the present paper concentrate on two topics: how the presence of natural cirrus changes the number of contrail ice crystals nucleated and how it changes the fraction of contrail crystals that survive the wake-vortex regime of the contrail evolution. Despite what the paper's title and abstract suggest, neither of these processes are actually simulated in the present work. The ICON-LEM simulations are employed purely to provide large and hopefully representative sets of sample atmospheric conditions with different levels of natural cirrus (which, on its own, is a potentially useful approach). Existing parameterizations taken from the literature for contrail nucleation and crystal loss are then evaluated on these sets of conditions. In the vast bulk of these cases the results of the cirrus on contrail nucleation and crystal loss are found to be essentially insignificant. This is not a new result; Gierens (2012) earlier reached the same results in more succinct and robust fashion. The authors here highlight the differences for the less common case of very heavy cirrus. Unfortunately, the parameterizations relied on here for contrail nucleation and crystal loss were not designed for, or tested on, the case of contrails forming within natural cirrus. The authors' extensions of these parameterizations to apply to this case leave out critical physics that is involved in these processes. In my opinion this makes the new conclusions drawn regarding changes in nucleation and crystal loss due to heavy natural cirrus untrustworthy and, in some regimes, even of the wrong sign. Nor are the results critically examined within the context that is really of interest here: how the natural cirrus might affect the potential impact of contrails on climate.*

*The text is generally clearly written, but verbose. Several statements are repeated in multiple places within the body of the paper. Further, much of the text is taken up paraphrasing old results from elsewhere rather than providing more explicit specifications of the present study.*

→Thank you for the in-depth review of the paper and for the many good suggestions. We have introduced many changes to our contrail scheme following your suggestions and reanalyzed the results. Our new results confirm many of your comments and we believe the paper was significantly improved.

We have modified the title and abstract in order to make sure that our approach, that is applying existing parametrizations in strongly differing atmospheric conditions and estimating the impact of the cirrus ice crystals by modifications to the parameterizations, is clear from the very beginning. We have rewritten large parts of the paper, describing the changed approach and results. We have not shortened the methods section including the descriptions of the parameterizations, on which our work is based, because we believe they are necessary in order to understand our approach and results. After all, we want our paper to be readable also for people that are not specialized in contrail modelling. Nevertheless, we have rewritten large parts of the paper and shortened the text in a number of sections.

We agree that most of the time changes in ice nucleation and ice crystal survival due to the presence of cirrus ice crystals are insignificant but we believe that this does not mean that the effects are not important. First of all, the fraction of cirrus cloud volume that leads to significant changes in contrail formation depends on the definition of a cirrus cloud. Since we consider cirrus clouds down to a

cirrus ice water content of $10^{-11}$ kgm$^{-3}$, it can be expected that large parts of the cloud field do not have an impact on contrail formation. Likewise, most contrails that form dissolve within seconds or minutes but that does not mean that the formation of contrails is not important. Instead it means that the climate impact is composed of a few large events. The impact of cirrus ice crystals on contrail formation turns out to be particularly large in synoptic situations when the climate impact of contrails forming in cloud free air is large as well. This means that the contrails that form within cirrus and that are impacted by the presence of cirrus have a relatively large life time and possible a large climate impact as well. We expand on the study of Gierens (2012) by scanning many different atmospheric states in order to analyze if pre-existing cirrus ice crystals have the potential to be important and therefore should be included in a contrail parametrization. Cloud properties can vary strongly and using typical values, as done by Gierens, cannot cover the whole breadth of possible effects.

*Specific points:*

*(1) In assessing the impact of existing cirrus on contrail nucleation the authors only consider the contribution of cirrus ice sublimated within the jet engines. The effects of cirrus ice that is mixed into the jet plume later are not considered. Depending on the temperature of the diluting jet at the time, these crystals might sublimate (providing an extra moisture source) or grow (providing a moisture sink). The authors use the change in contrail formation threshold temperature (T_sa), occurring through a change in the slope of the mixing line G in eqn. (1), as their primary metric of cirrus impact. This approach is already problematic in the presence of significant cirrus. The usual mixing line analysis and computation of T_sa relies on conservation of the water vapor during the mixing process (so that G is constant). But with the existing cirrus it is only the total water that is constant in the mixing process, not the vapor portion alone, because of growth and sublimation of the cirrus crystals. Furthermore, the contrail nucleation parameterization employed (Karcher et al., 2015) is based around the approximation that all droplets form at the same instant, and so the effects of existing ice crystals competing for available moisture during the nucleation process are not included. A recent LES study of contrail formation (Lewellen 2020) has shown that competition between ice crystals that form earlier and ones trying to form later can, in some regimes, significantly reduce the number of contrail ice crystals that would otherwise be produced. While for thin natural cirrus this effect on contrail nucleation should be negligible (as is the sublimation contribution considered by the authors), where the authors are reporting a significant cirrus effect (i.e., for very heavy natural cirrus and/or near contrail threshold temperatures) it need not be. This neglected contribution could lead to the presence of the cirrus sometimes reducing contrail ice nucleation, rather than always increasing it as concluded by the authors.*

→We have changed our representation of contrail ice nucleation to consider the cirrus ice crystals mixed into the aircraft plume and their sublimation or the deposition depending on the ice saturation ratio. We added an appendix describing how we estimate the sublimation/deposition on the cirrus ice crystals mixed into the plume based on the diffusional growth equation. Figures and text were modified throughout the paper.

We estimate that in some instances the impact of the preexisting cirrus ice crystals is to reduce contrail ice nucleation while in most occasions ice nucleation is enhanced. The likelihood of this happening depends on the atmospheric conditions. We find that ice nucleation on the 26[th] April in the upper levels where the contrail formation threshold is well exceeded the likelihood of reduced ice nucleation is around 20%, which is larger than at lower levels where atmospheric conditions are close to the formation threshold and larger than in the case of the thin cirrus section 3.2, line 517 to 521. In both

synoptic situations that we study and in both height levels reductions in ice nucleation are a few orders of magnitude smaller than the increases.

It is true that when considering the sublimation and deposition of cirrus ice crystals the usual mixing line approach is not fulfilled since the slope G would evolve in time. We calculated the slope of the curve describing the temporal evolution of G at the time of aerosol activation and compare with the mean slope when modifying the mixing line according to equation 3. We find that differences are below 1% which agrees with Gierens (2012) who found that the reduction in supersaturation due to deposition is far smaller than the production of supersaturation due to mixing. The largest changes to the plume's water vapor content due to sublimation / deposition happen very early in the plume life time so that treating the sublimation/deposition similar to the emission due to the combustion of fuel is a reasonable approximation. We added text accordingly right after equation 3.

We have mentioned the uncertainty coming from the assumption that all aerosols activate at the same time within the nucleation parameterization in several places:

In the introduction ' The number of ice crystals nucleated during contrail formation depends on the thermodynamic state of the ambient atmosphere and on aircraft and fuel parameters, in particular the number of aerosol particles released by the engine (Kärcher et al., 2015) but also on aerosol properties (Kärcher et al., 1998) and variable aerosol properties and inhomogeneities within the plume leading to ice crystals nucleating successively which has an impact on plume relative humidity and acts to decrease ice nucleation (Lewellen, 2020).'

In section 2.2.1 'All aerosols are assumed to activate and form droplets at the same time, $t_{act}$, called the "activation-relaxation time" neglecting the fact that aerosols that activate slightly earlier would have an impact on the plume relative humidity. This impact can sometimes be large in particular for large aerosol emissions (Lewellen, 2020).'

We have added a section on remaining uncertainties in the conclusions that mentions uncertainties coming from the assumption that all aerosols activate at the same time in the nucleation parameterization together with other sources of uncertainty (starting from line 769).

*(2) In extending the parameterization for the fraction of contrail crystals surviving the wake-vortex regime developed by Unterstrasser (2016) to include the effects of existing natural cirrus, the authors again only consider the cirrus crystals as a moisture source. Furthermore, they ignore the Kelvin effect in their implementation (e.g., in lines 283-285). This is a problem because it has been explicitly demonstrated in LES studies that crystal losses in this regime are significantly greater when the Kelvin effect is included than when it is not (see e.g., Lewellen et al 2014). The reason is that a significant portion of the crystal loss in the plume occurs in a regime where the water vapor pressure equilibrates to conditions which are subsaturated for the smaller crystals but not for the larger ones due to the Kelvin effect. Indeed it has been shown in exact analytic solutions of ice crystal populations under different conditions (Lewellen 2012) that significant crystal loss can occur even if the overall ice mass is slowly growing in time: the larger crystals grow at the expense of the smaller ones (a process known in more general contexts as ``Ostwald ripening''). In the present application this dynamic could lead to greater losses of contrail crystals in the presence of the larger natural cirrus crystals rather than reduced losses, including losses in the secondary wake where no adiabatic heating is occurring. Again, this effect on the contrail (like those included by the authors) should only negligibly be affected by the presence of thin cirrus, but for conditions where they are reporting significant cirrus effects (i.e., high cirrus ice number concentrations and IWC), this Kelvin-dependent process could even*

*change the direction of their reported effect, decreasing rather than increasing contrail crystal survival fractions.*

→We changed our parameterization in several ways. We included the diffusional growth phase after ice nucleation and before vortex descent in our analysis of the impact of cirrus ice crystals on the survival in the vortex phase. We calculate the competition of cirrus and contrail ice crystals after nucleation and before the start of the vortex descent using the whole diffusional growth equation. Once the vortex is descending we use again the whole diffusional growth equation to estimate the cirrus and contrail ice sublimation. We set the relative humidity to a constant slight subsaturation, choosing a saturation ratio of 0.98, a value that is often found in the descending vortices (personal communication Simon Unterstrasser). While we change the implementation of the parameterization of the vortex phase loss in our model we found an error in the earlier implementation. The survival fractions and in particular the changes in the survival fraction are now significantly reduced compared to our earlier estimates. Nevertheless, we find that in cirrus that has a large water content and additionally cirrus ice crystals are large the survival fraction is reduced and when instead the cirrus ice crystals are small then the survival fraction is increased. We changed figures and text accordingly.

*(3) The authors report greater relative changes in contrail ice crystal numbers due to natural cirrus for near-contrail-threshold conditions, largely because fewer crystals are nucleated there. It must be noted, however, that contrail ice numbers prove highly sensitive to a large host of variables near threshold conditions (see e.g., the simulation results in Lewellen (2020)). As a result, simple parameterizations in this regime, including both the nucleation and crystal-loss parameterizations that the authors are relying on for their conclusions, are much less trustworthy near-threshold than in most other regions of parameter space.*

-->We have added a sentence in the conclusions 'Furthermore, contrail formation close to the formation threshold is connected with a larger uncertainty than contrail formation far away from the threshold since details in the plume development may have a large impact, leading to varying ice crystal numbers resulting from slightly different LES simulations (Lewellen, 2020).'

*(4) In this work the authors only consider the changes in contrail ice crystal numbers due to the presence of the cirrus, not the changes in cirrus due to the passage of the aircraft (e.g., in metrics like equation (4)). For assessing the impact of aircraft on climate (presumably the ultimate motivation here) what matters is the net effect on the total system of contrail plus natural cirrus. For heavy cirrus or in near-contrail-threshold conditions where the authors claim significant increases in contrail ice crystal number, to what extent are these increases offset by losses in natural cirrus in the hot jets and descending wake? Further, the methodology for conducting the ``with'' vs ``without'' cirrus comparison is never explicitly defined, e.g., by specifying what variables are held fixed in the comparison. For example it is never actually stated whether it is water vapor or total water that is held fixed in the comparison. Given the results it is presumably the former, but the latter choice would in some sense give a more robust comparison (since water vapor changes more in time as the natural cirrus ages).*

→We do consider the loss of cirrus ice crystals due to sublimation in the engine. We do not consider the loss of cirrus ice crystals that are mixed in right after emission and that may be lost until plume saturation is reached since we do not have information on the size distribution of the cirrus ice crystals. We have now added a figure on the change in ice crystal concentrations due to contrail formation versus the original cirrus ice crystal concentrations (Figure 11). Changes in ice crystal concentrations are between 2 and 4 orders of magnitude larger than the original cirrus ice crystal

concentrations. That means that even if many cirrus ice crystals within the plume would additionally sublimate then the error in our estimate of the change in ice crystal number concentration due to contrail formation would not show up in the figure. But since reductions in ice crystal numbers due to contrail formation are much smaller and the error may possibly of a similar order of magnitude we refrain from showing the negative changes.

In lines 341-343 we specify now that we keep water vapor fixed when estimating the impact of cirrus ice crystals on contrail ice crystal loss. It depends on the question that you are asking whether water vapor or total water needs to be fixed. Keeping total water fixed our question would have been 'does the phase of the water in the grid box matter when we calculate contrail formation?' I believe that the answer is clear – it does matter. We instead ask the question 'Is the dependence of contrail formation on atmospheric variables unchanged even if ice crystals preexist in the background?'.

It is true that the ultimate motivation of our research is to estimate the climate impact of aviation induced cirrus perturbations. In this study we make the first step studying the contrail formation phase that is important for simulating the microphysical processes and the evolution of the contrail properties. In our future work we plan to study the evolution of those cirrus perturbations and their impact on optical properties.

(5) *It seems to me that the regimes where the current work concludes that effects of existing cirrus on contrail properties may be significant (e.g., heavy cirrus or near-threshold conditions) are not in fact ones where contrail climate impacts are potentially very significant. But how different cirrus scenarios might affect contrail radiative impact or longevity are never addressed in the paper. Radiative effects of contrails are clearly of potential concern when they seed significant, long-lived contrail cirrus in otherwise clear skies (i.e., where natural cirrus is optically thin or absent). On the other hand, the net radiative impact of a contrail shrouded within optically dense natural cirrus will naturally be expected to be much less (as well as occurring much less frequently). Likewise near-threshold contrails can be expected to have less impact because they occur less frequently, have fewer ice crystals and tend to be shorter-lived.*

→We agree that there are many questions and studies that can and should build on the work presented in this paper. We believe that any discussion of if or when contrail induced cloud perturbations are important for climate would be highly speculative. We have mentioned in our conclusions that the presence of cirrus impacts cloud properties in the outflow of conveyor belts and that those areas often support cloud for a very long time. That may point at cloud perturbations having a large impact simply due to their long life time. But this is speculation as well.

The aim of this paper was to introduce a contrail parameterization within ICON-LEM and study the contrail formation within preexisting cirrus, its variability and the impact of cirrus ice crystals on the formation.

---

## Author Comment (AC6)

**Reviewer #2 Comments (italicized), and responses (blue):**

*The paper addresses the question of contrail formation within cirrus clouds. Contrails constitute an important part of aviation impact on the atmosphere but the evaluation of their radiative forcing remains difficult to determine because of limited scientific understanding and complexity of the different processes to take into account. This paper is focused on the formation of "real life"-contrails, that do not only form in clear sky but also within existing clouds and we expect to know whether the effects are negligible or not. Therefore the wo rk provides valuable effort in order to better understand the processes and evaluate the effects. It uses a model including a cloud microphysical model and study the influence of contrails/cirrus clouds on each one for two synoptic situations. The paper focusses on the potential effects but is not meant to evaluate since it would require proper integration in a climate model. As a matter of fact, it is difficult to draw your own conclusions after reading whether preexisting cirrus clouds are important for contrail impact since the effects depend on the synoptic situation, the properties of cirrus clouds, the soot EI, the way the processes are treated etc. But it is important to go forward with such a work. It is perfectly in the scope of ACP.*

*The paper is a bit long, some details are repeated several times, and the paper is not very easy to read through, especially because within a paragraph you may have to refer to a lot of figures at different locations (see for example section 3.3.1 where figures 4b, 7a, 2, 3 are called in the first 10 lines).*

*A list of acronyms could also be useful to the reader since a lot of them are used in the text, sometimes unnecessarily (RBF or CWP are only mentioned once for instance).*

→Thank you for the review of our paper. We changed our paper according to your suggestions.

In the process of revising the paper according to both reviewer's comments we included a number of additional processes that have implications for the contrail ice nucleation and for the contrail ice crystal survival in the vortex phase. In particular, we included the impact of cirrus ice crystals that get mixed into the aircraft plume in the first second of the plume's lifetime before contrail ice nucleation section 2.2.2. Furthermore, we include now the diffusional growth phase of the plume ice water content after contrail ice nucleation and before vortex descent in our estimate of the impact of cirrus ice crystals on the contrail ice crystal loss within the vortex phase (section 2.2.4). For this estimate we use now the full diffusional growth equation (Appendix A). Finally, we found an error in our implementation of the parameterization of ice crystal loss in the vortex phase that we corrected. All together we find now that the impact of cirrus ice crystals on contrail ice nucleation is larger than estimated before and that the impact of cirrus ice on the vortex phase loss is most of the time negligible except for the parts of the natural cirrus that include many small ice crystals and those where contrail ice nucleation is very low and the cirrus ice crystal radius relatively small (section 3.3).

Accordingly, large parts of the paper were changed. In the context of the revision the text was shortened in many places in order to avoid repetition and we improved our use of abbreviations.

*Specific comments/remarks*

*The soot emission index that was used in the simulation, 2.5xl0$^{15}$ particles per kg fuel was fairly high. It is indeed mentioned that this is for the soot-rich regime. Recent works on the non-volatile particle emission certification process have also emphasized the importance of particles loss in the measurement system, and that could lead to underestimates of soot emissions, especially from previous studies and in flight. Surely the choice of this index has some consequences on the results and the sensitivity study of the survival fraction to the soot emission index that has been performed in section 3.3.1 should have been more emphasized (or advertised at the beginning of the paper as the reader may think that only one soot emission case has been treated).*

→ The soot number emission index of $2*10^{15}$ kg-fuel$^{-1}$ was chosen according to Bräuer et al. (2021) but we agree that this value is very uncertain and highly variable. We mention this uncertainty now in the conclusions. When choosing a lower soot number emission index, changes in ice nucleation due to the impact of cirrus ice crystals would remain the same for the lower soot number emission index except that the new $AEI_i$ would be limited to the lower soot number emission index. In the analysis of the impact of cirrus ice crystals on the survival of contrail ice crystals the soot number emission index and the number of nucleated contrail ice crystals is very important so that we have analyzed the sensitivity of our results to a range of soot number emission indices. We mention this sensitivity analysis now earlier in the paper e.g. in the last sentence of the introduction and in the last sentence of section 2.3.

*Regarding the effect of ice crystals sucked into the engine, Gierens (2012) indicates that the change in the water vapour emission index is very small, of the order of its variability regarding the fuel's composition. In this work, the effect depends on the airto-fuel ratio and the cirrus cloud ice water content (eq. 2). It would be nice to have at this point some example values so that the reader can figure out how much water vapour can be added to the plume. Besides, eq. 2 give an upper limit to the added H20 since it may be affected while going through the engine, including the hot combustion chamber.*

→In Gierens (2012), as well as in our paper, the change in the water vapor 'emission index' is dependent on the air to fuel ratio and the cirrus ice water content. We choose the same air to fuel ratio and the same water vapor emission index (1.25 kg per kg-fuel) as Gierens (2012). The cirrus cloud properties that we use are displayed in our figure 2. Our results are consistent with Gierens (2012) regarding the impact of the sublimation of cirrus ice crystals in the engine. Our figure 3 (dashed lines) shows that the sublimation of cirrus ice crystals within the engine amounts to maximally a few percent of the 'aviation induced change in the plume water vapor' that is the sum of the water emission due to fuel combustion and the sublimation of and deposition on cirrus ice crystals. For our thin cirrus we get a maximum change in water vapor 'emission' of about 0.2% (blue lines) and for our frontal cirrus day a change of about 4% (red lines) at a probability of about $10^{-2}$, which agrees with the 'order of percent' result of Gierens. The sublimation and deposition on cirrus ice crystals that were mixed into the plume after emission can occasionally be larger than the sublimation of cirrus ice crystals within the engine but it can be also negative. Gierens did not consider the impact of cirrus ice crystals that are mixed into the plume on the water vapor 'emission index'. Therefore, the maximum increase in the water vapor 'emission' due to the sublimation /deposition of cirrus ice crystals (figure 3, solid lines) is larger than the estimate of Gierens and, when assuming an IWC that would be representative of a relatively thick in-situ cirrus, can also be larger than the variation in the water vapor emission index given by IPCC. these results are described in section 3.2.

We have added a sentence in 2.2.2 shortly after equation 2 saying that the impact of cirrus ice crystals on the aviation induced water vapor content of the plume is discussed in section 3.2. We have also added a sentence in section 3.2 saying that we use the cloud properties as displayed in figure 2 for our calculations. Furthermore, we mention in the figure caption of figure 3 that the water vapor emission index is 1.25 and that the cloud properties are displayed in figure 2. In the summary and comparison to literature we compare our results to Gierens as explained above.

To our knowledge the $H_2O$ content of the air is not changed when going through the engine. Water does not accumulate in the engine.

*Please add some justifications on the choice of a "rough(ly) estimate" in equation 5: no Kelvin effect (are particles large enough at this point to exclude it?), spherical particles (correct for young contrails but not for cirrus clouds).*

→We use now the full diffusional growth equation (including the Kelvin effect) for calculating the diffusional growth of mixed-in cirrus ice crystals before nucleation and the growth of contrail and cirrus ice crystals after nucleation and when estimating the cirrus and contrail ice sublimation during vortex descent.  Please see our new Appendix A for details.

We assume spherical particles as we have no information on the particle shape or the mix of particle shapes of the cirrus ice crystals from our model. When estimating the impact of cirrus ice crystals that are mixed into the plume on ice nucleation and on the survival of contrail ice crystals we find that it is mainly cirrus containing very many small ice crystals that have the largest impact. This cirrus is connected with the frontal system crossing Germany on that day and has properties representative of a medium thick in-situ formed cirrus and partly a liquid origin cirrus (Krämer et al., 2020). Those ice crystals are likely close to spherical so that the assumption is likely a good one. Nevertheless, when cirrus ice crystals are larger our assumption of spherical particles may often underestimate the real sublimation and deposition on the mixed in cirrus ice crystals. Nevertheless, the changes in contrail ice nucleation and survival in the vortex phase are so much lower when cirrus ice crystals are large that we do not expect our results to change significantly even if we would have more information on the shape of cirrus ice crystals.

We have added a few sentences in the new appendix just after the introduction of the diffusional growth equation Line 819 to 824. We have also mentioned the dependency of our results on the shape as one of the uncertainties in our analysis (conclusion, line 769).

*Following point (3) in the summary, from line 627, the role of the temperature change on the saturation vapour pressure and in turn on the relative humidity is described. The sublimation of cirrus ice crystals and contrail crystals releases water vapour so that the system tries to reach a vapour/ice equilibrium. The change in the survival rate due to cirrus ice crystal sublimation should be considered a maximum since (if) cirrus crystals are treated the same way as contrail ice crystals (spheres, no influence of size). The competition between contrail and cirrus cloud particles during sublimation seems to be the important part and should be treated with the maximum accuracy.*

→As mentioned in the above points we have changed our parameterization and use now the full diffusional growth equation. Please see our new Appendix A for details.

As argued above, the impact of cirrus ice crystals on contrail ice nucleation and on the survival of contrail ice crystals during vortex descent is largest when the cirrus consists of many relatively small ice crystals. In this situation the assumption of spheres may not be that bad. If cirrus ice crystals deviate significantly from spheres then the deposition on the ice crystals would be larger as well as the sublimation would be faster during vortex descent. It is not clear to us if this would have mostly a positive or negative impact on contrail ice crystal survival. Since it is mainly large ice crystals that have shapes that are far from spherical and since large ice crystals usually lead to a very slight change in the survival fraction, we believe that the impact of assuming different shapes for large ice crystals would not be very large. But we agree that there is an uncertainty connected with our assumption that we can treat ice crystals as spheres. As mentioned in the answer to the previous point, we have added in the appendix a few sentences about the problem treating the impact of particle habit. We also added in the conclusions that the shape of ice crystals and its influence on the diffusional growth is one of the uncertainties that should be in future tested with LES modelling.

*In the conclusion section line 706, the authors conclude that "the pre-existing cirrus can lead to changes in the contrail formation criterion and, therefore, can lead to contrail formation when otherwise none would have formed". This is a strong statement for someone who would only read the conclusions, regarding to the text for instance statement line 607 "In large parts of the cirrus cloud field the presence of cirrus does not impact the contrail formation criterion and contrail ice nucleation significantly".*

→We have rewritten the conclusions and the statements should sound now less contradictory. In large parts of the cirrus clouds, the cirrus ice crystals have little impact on contrail formation except when IWC and ice crystal number concentration is high or if contrail formation is happening very close to the contrail formation threshold.

*Line 708-710 seems not so clear to me. "That means that the pre-existing cirrus ice crystals can lead to contrail formation in cases when otherwise the passage of an airplane would have dissolved the cirrus". Does this refer to taking into account or not the effect of pre-existing cirrus in the contrail formation process?*

→Yes, when we take them into account during contrail formation. We have reformulated the sentence and hope it is now less ambiguous. 'The presence of pre-existing cirrus ice crystals leads often to an increase in the contrail formation threshold temperature. Therefore, contrails may form and cause locally relatively high ice crystal number concentrations when, in the absence of the pre-existing ice crystals, no contrails would form so that only the sublimation of cirrus ice crystal in the engine would lead to a change in the ice crystal concentrations.'

*Sometimes "forgotten" in plume microphysical model studies for convenience or for the sake of simplification, ambient aerosols and especially ice cirrus clouds could be taken into account for a more detailed sensitivity analysis. It could be one more recommendation of the conclusion.*

→We do analyse the impact of cirrus clouds on the contrail ice nucleation and the survival of ice crystals in the vortex phase. We do not consider the full impact of ambient aerosols on contrail ice nucleation. While prescribing soot-rich emissions the ambient aerosols do not play an important role in determining ice nucleation unless their properties are significantly different from the soot properties which could lead to larger errors when using the contrail ice nucleation parameterization of Kärcher et al (2015). We added in the conclusions that variability in aerosol properties together with inhomogeneities have an impact on ice nucleation and constitute an uncertainty of the parameterization of contrail ice nucleation and therefore of our approach.

Technical corrections

*Line 34 "since in IPCC style double C02..."*

It is not clear to us what this comment refers to.

*Line 146 "vertically consistently"*

done

*Figure 2 (a), (b), (c), typo in the vertical axis legend "Tamperature"*

done

*Figure 4, in the caption unit hPA instead of hPa in (~250 hPA)*

done

---

## Author Response (AR2)

Answers to reviewer I:

We appreciate the comments of the reviewer. In the following our answers to the reviewer's concerns are marked in blue while the reviewer comments are in black.

This revised manuscript is substantially improved in several respects: it is now clearer from the text what is being done; some of the parameterizations are improved somewhat; and some important shortcomings/uncertainties are now acknowledged. Even with these changes, however, the primary concern from my original review largely remains: the approximations and analysis being used are not at the level required to justify many of the findings that are being emphasized. I would strongly encourage the authors to focus on results that their analysis defensibly supports. The manuscript could also benefit from a tightening up of the prose throughout.

The aim of this research -- addressing the impact of pre-existing cirrus on contrail formation and properties -- is a useful one. Further, the approach employed and results presented are, I think, sufficient to draw a valuable conclusion toward that aim: that the presence of natural cirrus is unlikely to significantly increase the climate effects of contrails. To me this is the natural conclusion to be drawn from e.g., fig.3, which shows enhancement of water vapor of at most 20%, and at that level for only a tiny fraction of the cirrus samples (1 in 10000); or from figs 5,6 that show for large contrail ice numbers (i.e., where contrails have a potential for significant impacts) that Delta\_n/n is quite small almost everywhere in the sampling of cirrus conditions.

Further, even the modest enhancements of contrails by cirrus that are seen in the results may be reasonably interpreted as upper bounds, since the shortcomings in the parameterizations and analysis employed (e.g., those listed in my prior review) generally tend toward overestimation of contrail ice crystal number.

This general result is admittedly one-sided: because the radiative effects of the cirrus have not been considered in this work, the results do not rule out the possibility (I would suggest likelihood) that in some regimes the presence of the cirrus might significantly reduce the potential climate impact of contrails.

Unfortunately, the authors seem reluctant to simply focus on what is essentially a reinforcement of the conclusions of Gierens (2012). They do make statements in support of these conclusions in several places, but elsewhere include contradictory statements stressing smaller effects in the tails of their distributions to try to argue for a significant strengthening effect of cirrus on contrails (e.g., in lines 22-23, 697-700, 702-704, 706-707, 713-714, 755-757). If the authors want to argue for the validity of results at the few percent level (or, in many cases in the paper, fractions of a percent), they need to demonstrate that their modeling uncertainties in the regimes in question are actually less than this level. Given the parameterizations used and the regimes involved (heavy cirrus and/or near-threshold conditions) their current model/methods would fail this test. The problem is exacerbated by trying to argue the general significance of results that are non-negligible in only a tiny fraction of the author's cirrus samples. And there are statements throughout the manuscript along the lines of ``the effect is larger in case X than in case Y'' where the effects in both cases are negligible relative to the uncertainty level in the model itself.

We disagree with the reviewer on a large number of points. In particular we disagree with the conclusion 1. 'the presence of natural cirrus is unlikely to significantly increase the climate effects of contrails', 2. that the low fraction of cloudy grid boxes that have a large impact is an indication that the processes have no significant impact on climate and 3. that the increase in plume water vapor of up to 20% means that the climate impact is negligible and 4. that Delta n/n is quite small almost everywhere is a sign for a low impact on radiation.

In this paper, we do not want to draw conclusions about the climate impact of contrail formation within cirrus, as the reviewer would like us to, because there are many other factors that would need to be considered within the calculation. Our goal is to clarify the processes that need to be included in such a study.

**1. Radiative impact**

One of the reviewer's main arguments is that the effects that we are studying do not have an impact on contrail radiative forcing. We disagree. As we said in the last review, our results do not support this conclusion, nor do they support the opposite. The effects may well have a climate impact. But the estimation of this climate impact goes far beyond the simple arguments of the reviewer and should be part of a subsequent study.

Our disagreement is caused by the fact that the reviewer assumes that the radiative forcing due to contrail perturbations is necessarily positive (if contrail ice crystal numbers are increased) and that contrail radiative forcing is on average proportional to contrail ice crystal numbers after the vortex phase. In the case of contrails forming in cloud-free ice-supersaturated air this is the case. But it is not true in the case of contrails forming within pre-existing cirrus. The radiative impact of contrails forming inside cirrus depends on the optical depth perturbation caused by contrail formation and on the optical depth of the undisturbed cloud. Among others, Meerkötter et al. (1999) and Markowicz and Witek (2011) discuss the dependence of radiative forcing on optical depth and show a decrease in contrail warming once a certain optical depth is exceeded. Therefore, a large number of contrail ice crystals after the vortex phase (and a large correction of contrail ice crystal numbers due to cirrus ice crystals) may lead in an optically thin cloud to an enhanced warming and in an optically thicker cloud to a cooling relative to the undisturbed cloud. The overall impact of contrail formation within cirrus likely depends on the balance of the two effects. A large optical depth perturbation in a cirrus that has an optical depth that causes the maximum radiative impact may well have a smaller radiative impact than a relatively small optical depth perturbation at a much lower or much higher cirrus optical depth. Furthermore, the change in radiative forcing due to the contrail perturbations depends on the ice crystal habit of the natural cloud and their change due to contrail formation.

Since corrections in ice nucleation are large in clouds with large ice water content which are likely connected with large optical depth, we may possibly underestimate the cooling of the contrail perturbations when disregarding the impact of cirrus on contrail formation. The reviewer says that 'the presence of natural cirrus is unlikely to significantly increase the climate effects of contrails.' As the reviewer suggests, there is actually a good possibility that the impact of the cirrus ice crystals on contrail formation is decreasing the climate effects of contrails, although it appears from the previous review that we disagree on the reasons for this cooling impact. Disregarding the impact of cirrus on contrail formation may then result in an overestimation of the contrail impact. Determining this climate impact should be the topic of future research.

We have added text in the conclusions discussing the difficulties assessing a potential climate impact of contrail induced cirrus perturbations even though this is not the topic of our paper.

2. Statistics

The reviewer states that the low fraction of cloudy grid boxes where the impact is large is an indication that the processes have no significant impact on climate. We do not believe that this is a valid argument because the places in which the impact of cirrus on contrail formation is larges are the places in which contrails can be expected to have a large impact. This means that there are reasons to expect that the impact of cirrus ice crystals on contrail formation can have a significant effect.

The formation of contrails in cloud free air is known to have a large impact in large scale ice supersaturated areas connected with frontal systems (Bier et al., 2017; Burkhardt et al., 2018) which is the same regime where we find an impact of cirrus ice crystals on contrail formation. In particular, large scale upward motion and the connected water transport into the upper troposphere, as found in the context of frontal activity and conveyor belts, facilitate a large radiative impact. This means that the regimes in which the corrections in contrail ice nucleation are relatively large are the same regimes/ times when contrails formed in cloud free air often have a large climate impact. This points at the possibility of cirrus ice crystals having a significant impact on contrail properties within cirrus and leading to a significant radiative impact.

Furthermore, it is well known that a small fraction of contrails forming in cloud free air explain a large share of the climate impact of contrail cirrus. This means we need to study contrail formation within those regimes where the climate impact is strong even if those situations are not common. Studying average contrail properties or the average impact of cirrus on contrail formation may be irrelevant (the same way as it is close to irrelevant to know the average life time of contrails). Our study agrees in so far with the Gierens (2012) study that on average there is hardly any impact of cirrus on contrail formation. But it is the whole point of the paper to probe a large number of regimes, atmospheric states and cloud properties in order to find out if in some cases it does matter. If those situations, in which the impact of cirrus ice crystals on contrail formation are relatively big, are situations in which the climate impact may be large, then this warrants the inclusion of this paper to study the climate impact of the cirrus induced changes. This climate impact should be studied in a subsequent study (see above).

The argument of the reviewer is not really clear here 'If the authors want to argue for the validity of results at the few percent level (or, in many cases in the paper, fractions of a percent)'. In case this means that the reviewer is doubting that there is a signal at all given that the fraction of larger changes is so low:

If we were conducting a field significance test, then a significant result in a tiny fraction of grid boxes should be expected simply by chance and would be a sign that there is no physical connection. But we don't do that. In our case a small fraction of large changes simply means that in certain conditions (mainly large ice water content IWC) we can see a signal. The physics behind this connection is clear. The low fraction of large changes is simply caused by the fact that we use IWC 10-11 kgm-3 as a cloud mask and therefore include subvisible cirrus. As we commented in our last revision, the probability of large changes is a function of the minimum IWC that we use as a cloud mask. By setting the threshold to a higher value, the fraction of large changes can be easily increased.

We have modified the text at the end of section 3.2 (line no 556 to 559).

**3. Change in plume water vapor negligible**

The reviewer says that the presence of natural cirrus is unlikely to significantly increase the climate effects of contrails because plume water vapor is enhanced by at most 20%. As our analysis shows, this increase in plume water vapor can lead to changes in the contrail formation threshold of a few Kelvins (maximum 2K) and associated increases in ice nucleation can have the same order of magnitude as the ice nucleation when neglecting the impact of cirrus ice crystals even at temperatures of several Kelvins below the formation threshold. We judge this to be a significant change.

The reviewer says that we should accept that our study reinforces the conclusions of Gierens (2012). Gierens estimates the enhancement of the plume water vapor to be on the order of a few percent in case of a thick cirrus while we show that it can be as large as 10% and very seldomly reaches 20%. As

we show in the paper, this difference is mainly due to the fact that Gierens (2012) does not consider the impact of cirrus ice crystals entrained into the plume while we do.

The discussion of this topic can be found in section 4 (line no 638-649).

**4. Delta n/n**

The reviewer argues 'for large contrail ice numbers (i.e., where contrails have a potential for significant impacts) Delta\_n/n is quite small almost everywhere in the sampling of cirrus conditions'. As we point out under 'climate impact', the relative change in itself is not an indicator for the climate impact of the cirrus induced changes. Instead the climate impact depends crucially on the background optical depth of the cirrus.

As discussed under 'Statistics' it is true that in large areas of the clouds sampled the change is small. But as we show in the paper it is not true for cirrus with a large ice water content. In figure 5 and 6 we show the contrail ice numbers,  $n_i$ , when not considering the impact of cirrus ice crystals on contrail ice nucleation and its change  $\Delta n_i$  when considering sublimation and deposition on cirrus ice crystals. Whereas on the main air traffic levels changes can reach maximally 10% of  $n_i$ , on the lower levels changes are larger and can be of similar size than  $n_i$ . Since the radiative impact is not a function of contrail ice crystal numbers alone (see above under 'Radiative Impact'), the analysis should not concentrate only on cases of very high contrail ice crystal numbers (~ 108 m-3).

Concerning your comment 'Further, even the modest enhancements of contrails by cirrus that are seen in the results may be reasonably interpreted as upper bounds, since the shortcomings in the parameterizations and analysis employed (e.g., those listed in my prior review) generally tend toward overestimation of contrail ice crystal number.'

We assume that the reviewer refers here mainly to his results from Lewellen (2020) who shows that box model estimates overestimate contrail ice crystal numbers when compared to LES for high soot number emission indices. Since we assume a soot number emission index of 2.5\*1015 kg-fuel-1 we are still in the regime where the reviewer judges the box model to agree with LES. Furthermore, the Kärcher et al. (2015) parameterization may generally overestimate ice nucleation relative to the box model estimates since it does not include subsequent nucleation. But, in particular, when contrails form within a cloud large differences in aerosol properties should be severely reduced, limiting this sensitivity (see below for more details).

Finally, the sensitivity of contrail ice nucleation on cirrus ice crystals is valid even if the parameterization of Kärcher et al. (2015) should generally overestimate contrail ice nucleation. An increase in plume water vapor is bound to change the contrail formation threshold and ice nucleation independent of the fact that the nucleation parameterization that we use does not consider e.g. the impact of plume inhomogeneities on ice nucleation. It is common in climate science to evaluate a systematic change in a field that is subject to large uncertainty and variability, which is often caused by limited information on e.g. aerosol concentrations and properties or connected with the low resolution of a model.

You indicated the following lines (22-23, 697-700, 702-704, 706-707, 713-714, 755-757) which highlight differences to Gierens (2012). As we argued above we find significant differences to Gierens and believe that it is appropriate to discuss those differences and believe that they can have an impact (see above). Please see below for the discussion of modeling uncertainties.

*Further comments in regards to the authors' responses to the specific numbered points in my original review (using the original numbering) follow:*

(1) On approximations in the nucleation treatment: Some improvements have been made here (both in the approximations used and in the caveats included in the text). Including some estimate of the effects of cirrus crystals mixed into the exhaust plume is an improvement, though it is quite rough (as is clear from appendix A). Whether these changes are adequate depends on the conclusions one is trying to draw (see discussion above). The changes made a sizable percentage difference in some of the results the authors are highlighting (e.g., in figs 3-6). If the remaining shortcomings (e.g., a single activation time, not including a size spectrum so crystal losses can't be assessed, neglecting effects of plume inhomogeneity, omitting some feedbacks, etc.) were addressed one could expect to change the results at least as much (i.e., the uncertainty level is sizable).

The reviewer comments that our estimates presented in appendix A are quite rough but does not hint at shortcomings. We assume here that he refers to the temporal evolution of the dilution rate. We use the dilution rate given by Kärcher et al. (2015) which is again based on Kärcher (1999). Kärcher (1999) shows that the entrainment rate used in Kärcher et al. (2015) is a good fit to the entrainment from LES (Gerz et al., 1998) at times >~0.01s. Since we start calculating entrainment and sublimation of cirrus ice crystals at 0.01s we should not overestimate sublimation due to overestimating entrainment. Another reason for the 'rough estimate' may be the impact of neglecting the ice crystal size distribution - please see below for a discussion. The error of our estimate of the ice sublimation/deposition when estimating from a midway value instead of calculating the temporal evolution is a few percent which is certainly a good estimate.

**Most of the appendix was rewritten.**

In the following the reviewer gives a list of processes that may or may not impact our simulations significantly with no indication of which of those processes the reviewer regards to be most important and why. Most of the processes (single activation time and plume inhomogeneity) may change the estimate of the overall number of contrail ice crystals but may change little in our estimate of the sensitivity of contrail ice nucleation to cirrus ice crystal sublimation/deposition within the plume.

**Plume inhomogeneity:**

The overestimation of contrail ice nucleation due to not resolving plume inhomogeneities we expect to be small, since the reviewer concludes in Lewellen (2020) that plume inhomogeneities would lead to changes in ice nucleation particularly when choosing very large soot number emission indices such as 1016 kg-fuel-1 or higher. For soot number emission indices of 1015 kg-fuel-1, which is close to what we chose, the reviewer concludes that box modelling produces a good estimate of ice nucleation.

*Text had been added during the last revision in section 2.2.1 and was now changed (line no. 227 to 230)..*

**Consecutive nucleation (single activation time):**

In the reviewer's paper (Lewellen, 2020) he concludes that incorporating differences in activation time due to different aerosol properties is of secondary importance, less important than plume inhomogeneities. Including different activation times requires information on e.g. concentration, size distribution and hygroscopicity of ambient and emitted aerosols that are not readily available. The reviewer chose in his paper very different values for hygroscopicity of ambient aerosols than Kärcher et al. (2015), maximizing the difference between emitted soot and ambient aerosols. Differences

appear to be caused by different assumptions in the chemical composition of the aerosols. The assumption of equal concentration of large and small aerosols within Lewellen (2020) is additionally likely to overestimate the impact of different aerosol properties.

The impact of consecutive nucleation is dependent on the entrainment of ambient aerosols and their properties. Within existing cirrus, we can exclude entrainment of aerosols into the plume that preferentially form ice crystals. But aerosols will be added to the plume due to the sublimation of ice crystals within the engine. When ice crystals sublimate within the engine the sulfuric acid droplets, an aerosol on which very many ice crystals form, evaporates while the few IN, such as soot or dust, may be released with unknown properties increasing the already high soot number emissions slightly. We conclude that the problem of sequential activation due to differences in aerosol properties is much reduced when estimating ice nucleation within preexisting cirrus.

We added text accordingly in section 2.2.1

**Not including a size spectrum so crystal losses can't be assessed**

We have performed some offline calculations trying to estimate the size and number of ice crystals that may sublimate completely within the plume. We find that if an ice crystal is mixed into the aircraft plume at about half the time between emission and ice saturation then ice crystals up to a radius of around 1.5  $\mu$ m can completely sublimate. Assuming a grid box average ice crystal radius of 6  $\mu$ m (5  $\mu$ m), which are the smallest values that we (only occasionally) find in our simulations, and assuming ice crystal sizes to be distributed according to a generalized gamma distribution (Seifert and Beheng, 2006), about 5% (10%) of ice crystals within the grid box have a size of below 1.5  $\mu$ m. From this calculation we conclude that the impact of the change in cirrus ice crystal numbers within the plume due to complete sublimation before nucleation is rather limited. Additionally, the complete sublimation of cirrus ice crystals would likely have a smaller impact on the sublimation than on the subsequently happening deposition and, therefore, would be likely to lead to an underestimation of the increase in plume water vapor and of the impact of cirrus ice crystals on contrail ice crystal nucleation. This means that, when considering the loss of ice crystals during sublimation on our estimate of sublimation and deposition on cirrus ice crystals, our estimate is very likely conservative.

**Large parts of the appendix were rewritten.**

It is certainly true that 3D LES are much better suited to simulating all the processes relevant to estimating ice nucleation. But our ICON-LEM is better suited to sampling a large number of different atmospheric conditions connected with certain synoptic situations and estimating the impact of the atmospheric development on the development of contrails. In order to do this the model needs to capture as much of the physics of contrail formation as possible even if the resulting estimates will have a larger uncertainty than LES estimates given one particular atmospheric state.

In summary, plume inhomogeneities and consecutive nucleation may affect the number of contrail ice crystals nucleating but may not have a significant impact on the sensitivity of contrail ice nucleation to sublimation and deposition on cirrus ice crystals. Errors in estimating sublimation and deposition on cirrus ice crystals within the plume, such as those errors connected with neglecting cirrus ice crystal loss, would have a direct impact on this sensitivity but we estimate that the relative error in our estimate is likely small.

(2) On approximations in the crystal loss parameterization: The changes implemented are in the right direction but still miss most of the important effects. The revised treatment is implemented as if the Kelvin dependent effects were important only early on, before the ``vortex descent''. This is not so. Through much of the vortex descent, conditions arise in which the larger crystals continue to grow

while the smallest sublimate away. Further, there is significant detrainment from the descending plume leaving portions that never descend enough to achieve subsaturated conditions for all crystal sizes (but with losses for the smallest). And to emphasize again: the parameterization of Unterstrasser (2016) is based empirically on LES results that properly include the Kelvin effect, but not with ambient cirrus crystals included. The current treatment in the manuscript is probably best interpreted as estimating an upper bound on the crystal survival fraction during vortex descent, perhaps greatly underestimating the potential in some heavy cirrus regimes for additional Kelvin dependent losses. Whether this treatment is adequate again depends on what conclusions one is trying to draw (see comments above). I would argue that the only conclusion in this regard that the results in the manuscript will currently reliably support is that crystal loss rates are unlikely to be significantly reduced by the presence of existing cirrus; I would recommend heavily trimming the lengthy section 3.3 accordingly.

The reviewer misunderstands our approach. We do not assume that the Kelvin effect ends at vortex descent and we do not simply use the parameterization of Unterstrasser (2016) once the vortex is descending. Instead we use the parameterization after adjusting the water vapor emission according to the sum of the deposition on cirrus ice crystals before the vortex phase descent and the sublimation during vortex descent. 'In order to be able to use the parameterization of Unterstrasser (2016), that does not include the impact of cirrus ice crystals on the survival fraction of contrail ice crystals, we adjust the water vapor 'emissions' of the air plane, which is an input in the parameterization. While the water vapor emission is usually given by the EI\_H2O coming from fuel combustion, in the context of contrail formation within cirrus we use the 'aviation induced increase in water vapor' that includes the sum of the sublimation of and deposition on cirrus ice crystals that changes the water vapor content of the plume. '

Therefore, we first calculate the ice crystal growth in the time between ice nucleation and vortex descent using the depositional growth equation to estimate the temporal evolution of the size of the contrail and cirrus ice crystals considering the Kelvin effect. We then estimate, using the full diffusional growth equation, how much of the cirrus ice water can sublimate in the time it takes the contrail ice crystals to sublimate in the descending vortex and limit the cirrus ice water sublimation using the time scale of vortex descent. Both those steps include the Kelvin effect. Our final step is applying the parameterization of Unterstrasser (2016) while adjusting the water vapor 'emission' according to the sum of the deposited and sublimated water vapor on the cirrus ice crystals. This means that we do consider the Kelvin effect before and during vortex descent.

This approach should give us a very rough estimate of the impact of cirrus ice crystals on the survival of contrail ice crystals within the vortex phase. Since, on the one hand, the time until vortex descent is not well defined and, on the other hand, during the first few seconds of vortex descent relative humidity may be such that cirrus ice crystals could grow at the cost of contrail ice crystals, we vary the growth time period before vortex descent (19s instead of 9s) in a sensitivity simulation. We hardly find a difference in the resulting impact of cirrus ice crystals on contrail ice survival. Deposition and sublimation rates close to ice saturation are very small and differences in cirrus and contrail ice crystal radii and cirrus ice crystal concentrations are not large enough to let cirrus ice crystals grow significantly at the cost of contrail ice crystals.

We have rewritten large parts of section 2.2.4 and hope that it is now easier to understand.

Finally, Yes, we agree with the reviewer that the presence of cirrus ice crystals does not significantly change the contrail ice crystal survival. We have cut section 3.3 significantly.

(3) On uncertainties in the near-threshold regime: The caveat added to the revised text is useful and important. But it does not alter the fact that the enhanced uncertainty in the near-threshold regime likely swamps most of the results that the authors are calling attention to there.

In the paper we call attention to the changes in ice nucleation that are connected with large cirrus ice water content independent on if a contrail forms close to the formation threshold or not. Besides, assuming that the reviewer uses the same terminology here as in his 2020 paper, a 'near threshold case' refers to contrail formation within a few tenth of a degree below the formation threshold. As can be seen in our figures 5 or 6 the impact of cirrus ice crystals on ice nucleation can be significant at temperatures several Kelvins below the formation threshold. Therefore, our main results do not come from near threshold cases (using the above definition of 'near threshold').

Furthermore, a systematic change in the contrail formation threshold, as we are finding due to the sublimation of cirrus ice crystals in some regimes, is certain to increase ice crystal numbers within the cirrus even if the resulting ice crystal numbers are highly sensitive to ambient temperature and the contrail formation threshold temperature, humidity, contrail inhomogeneities and many other variables. Therefore, the uncertainty does not 'swamp' the signal. Instead our signal is a systematic change of the highly variable ice nucleation.

Please note that it is common in climate science to evaluate a systematic change in a field that is subject to large uncertainty and variability.

(4) On not including the aircraft induced cirrus losses, etc: I find the added section 4 and fig. 11 (as well as related statements in various places such as lines 16-17, 530-534, 743-745, 767-768, 785-786) rather beside the point. The point in this work is not to compare ``contrail" with ``no contrail", but to compare ``contrail in presence of cirrus'' with ``contrail in absence of cirrus''. It is well known that in contrail-favoring conditions, contrails typically produce much greater crystal number densities than natural cirrus; the current work gives no new insights in this regard that I am aware of. Nor is fig.11 convincing evidence that losses of cirrus crystals due to the aircraft (which the authors are not computing) are necessarily negligible compared to the contrail ice numbers in the regimes where the authors are highlighting cirrus effects on contrails. In typical conditions within fig 11 it is indeed the case that the number concentrations from the contrail greatly exceed those from the cirrus. But in the small fraction of the cases that the authors tend to emphasize in the current presentation this needn't be so, either because the cirrus number concentrations are very high, or the contrail numbers are much lower (near-threshold). And I think the authors are missing an opportunity here in fig. 11. What they should compare for their purposes is this figure with the analogous one where the contrail numbers are those produced by their model when not including the existing cirrus. Judging from the results presented elsewhere in this paper, I suspect that the two figures would look nearly identical almost everywhere, providing good visual support for what I am arguing should be the main conclusion of this work.

It is true that the new section added a slightly different direction to the paper. Originally, we added this section due to the reviewer's comment 'For assessing the impact of aircraft on climate (presumably the ultimate motivation here) what matters is the net effect on the total system of contrail plus natural cirrus.' Apparently, we misinterpreted this comment.

Anyway, we cut this section since a proper discussion and evaluation of this would make this section too lengthy. We did not cut statements referring to the impact of contrail formation for cirrus cloudiness as long as those statements are supported by the results of the other sections. We believe that those statements are of benefit for a general reader. After all, we do not want our paper to be only read by the 'contrail community' but we would like it to be also informative for people generally interested in cloud processes who may not find those statement all that trivial. As to the description of the methodology for comparing ``contrail in presence of cirrus'' with ``contrail in absence of cirrus'' in this work: the added statement in lines 341-343 remains ambiguous. What comprises ``exactly the same situation except for the absence of cirrus ice crystals'' depends on what variables are being used to describe that situation (e.g., water vapor vs total water), and these have not been specified here. It is not possible to set the cirrus ice number (or IWC) to zero while leaving all other variables exactly the same.

The text reads now 'In section 3.3 we will discuss the impact of cirrus ice crystals on the loss of contrail ice crystals during the vortex phase comparing it to the ice crystal loss that the contrail would have experienced in the same situation except for the absence of cirrus ice crystals and with a corresponding reduction in total water.'

(5) On considering where contrail radiative impacts are likely to be high (and where not): My concern here remains as before (and essentially unacknowledged in the manuscript). The regimes where the authors are claiming ``significant'' enhancements to contrail ice numbers due to the presence of existing cirrus are precisely regimes where the potential climate impacts of contrails are likely to be naturally reduced: near-threshold conditions (where contrails will have lower optical depths and shorter lifetimes) or high-IWC, high-ice-number cirrus (where the surrounding cirrus with its large optical depth can largely mask the radiative effects of the embedded contrail, as well as reduce the contrail's crystal numbers over time through Kelvin-dependent crystal losses).

The answers to this point are already given above in our discussion of the 'radiative impact' and in connection with your comment 3.

**Newly added literatures:**

Markowicz, K. M., & Witek, M. L.: Simulations of Contrail Optical Properties and Radiative Forcing for Various Crystal Shapes. Journal of Applied Meteorology and Climatology, *50*(8), 1740–1755, https://doi.org/10.1175/2011jamc2618.1, 2011.

Meerkötter, R., Schumann, U., Doelling, D. R., Minnis, P., Nakajima, T., & Tsushima, Y.: Radiative forcing by contrails. Annales Geophysicae, 17(8), 1080–1094, https://doi.org/10.1007/s00585-999-1080-7, 1999.

---

## Author Response (AR3)

**Answers to editor**

Dear Martina,

We have now revised our paper again.

*After carefully studying the report, my decision is now to accept the manuscript with minor revisions (review by editor). In the revised version, I would ask you to take into account the recommendations of the referee, i.e. to tone down the interpretation of the results (see comments Category (A)) and to better discuss the inaccuracies of the used parameterizations (see comments Category (B)).*

In particular we have toned down our interpretation in the abstract and summary (as requested by reviewer 1 in Category A). We say now clearly that the effects are often negligible and describe the atmospheric conditions in which they are not.

We have also now added an additional estimate of the probability of significant changes in contrail formation when using a different IWC threshold for selecting the cirrus that we include in the analysis. This should give the reader a better quantitative idea of how uncommon large changes are. A better quantitative measure for the frequency of large changes should make the exact wording less of an issue.

Changes made in connection with Category B were mainly made in section 2.2.4. Unfortunately, we still believe that the reviewer did not understand how we estimated the impact of cirrus ice crystals on the vortex phase loss since he still insists that we did not consider the competition between cirrus and contrail ice crystals in the descending vortices. In order to improve the description and to make it easier to understand we rewrote section 2.2.4 and iterated it with the LES modeling group at our institute that specializes in modeling the contrail ice crystal loss in the vortex phase. We hope very much that the section is now generally easier to understand.

Nevertheless, we have also improved our description of the assumptions and uncertainty of our parameterization in the conclusions adding 'Furthermore, we assume a constant plume subsaturation within the descending vortex in order to calculate the competition between cirrus and contrail ice crystals and calculate contrail ice crystal survival after adjusting the plume water vapor content consistent with the impact of the cirrus ice crystals on plume water vapor. '

*Please provide a revised version of the paper together with a manuscript with changes tracked. Also, it would be good to briefly answer to the major points raised by the referee.*

We have answered all of the reviewer's comments. But some of the comments seemed to relate more to our answers to his comments and less to the paper or seemed to agree with parts of our argument. Those points we partly did not answer.

**Answers to comments of Reviewer 1**

*The authors have chosen not to follow most of the suggestions in my review of the prior revision of this manuscript (an exception is in the substantial pruning of section 3.3, which has improved the manuscript). The authors' attempts to rebut my concerns, both in additions to the manuscript and in replies to my review comments, I found unconvincing, and sometimes plainly incorrect. Accordingly, the bottom-line conclusion from my prior review remains unchanged: I think this paper needs revisions if it is to be published. While I consider these in the category of "major revisions" because they involve the primary conclusions, I am not requiring changes in the actual analysis or figures, just in making the text consistent with the results presented and the level of approximations represented in the modeling.*

*The paper's stated aim is to study the impact of pre-existing cirrus on contrail formation, in particular its effect on the number of contrail crystals that nucleate and their survival fraction in the vortex regime. As noted in detail in my prior reviews, my concerns about the manuscript can be grouped into two broad categories: (A) that the presentation, particularly in the abstract and conclusions, gives an inflated impression of the potential importance of these effects relative to what the results presented actually show; and (B) that the accuracy of the parameterizations employed for both nucleation and crystal loss are much rougher and more uncertain than the presentation implies. I will consider these categories in turn below, but confine my comments to changes made in the current manuscript version and authors' comments about them. I won't reiterate all the points in my last review -- not because I think they have been adequately addressed, but because I have no changes to make in them.*

*Category (A):*

*In the abstract the authors highlight the largest differences they find (e.g., "...which can be as large as 2K", "...contrail ice nucleation rates can be significantly increased...") without highlighting that these occur only in the rare tails of their simulation cases (less than a fraction of a percent). Nor is the frequency of occurrence accurately presented in the summary or conclusion sections. For example lines 663-664 ("We conclude that the sublimation of cirrus ice crystals in the engine and the impact of cirrus ice crystals mixed into the plume can have a significant impact on contrail formation.") are given with no indication of the rare occurrence. Or in lines 653-655, where the authors concede that the effects are not significant "in large parts of the cirrus cloud field", but their own figures indicate "large parts" is really "almost everywhere". I think finding these "cirrus correction effects" at a significant level (as measured by changes in crystal number) only rarely is strong evidence for the unlikeliness of these effects having a significant impact on the climatic effect of contrails. The authors in their comments and additions try to argue otherwise in a few ways, none of which I find convincing.*

Text passages in the abstract and in former lines 663-664 and lines 653-655 were changed.

*(A1) First, in their comments and the added lines 555-558, the authors try to argue that this is not the case because the rarity seen in their results is only an artifact of "the minimum IWC that we use as a cloud mask". This is a spurious argument. It is true that their choice for defining their probability distributions is arbitrary, but it is adequate for a rough picture. If they had normalized the probabilities with a more physically relevant choice such as fraction of all "conditions producing contrails" or "conditions producing contrails above some significance level" then the dependence on the choice of "minimum cirrus IWC" would drop out, but the fraction of cases with significant "cirrus correction effects" would still be very small (particularly since in assessing these effects for contrail-climate-impact purposes one should consider also all the contrails forming in clear skies). I suggested a possibly better way to present the relative importance of these effects in my last review by modifying the authors' then fig.11, but the authors chose simply to remove that figure instead.*

When normalizing with the fraction of conditions producing (significant) contrails we do not eliminate the dependence on the choice of minimum cirrus IWC. At low cirrus IWC, ice supersaturations can be high and contrail formation leads to large numbers of ice crystals similar to contrail formation in cloud free ice supersaturated areas. We now analyze the dependency of our results on the cirrus IWC. We have recalculated the probability of larger changes when considering only cirrus with ice water content higher than $10^{-5}$ kgm$^{-3}$ in order to quantify this dependency of our results on the cirrus IWC. We have added accordingly a few sentences in section 3.2.

*(A2) Second, in their comments and added lines 703-714 they try to argue that this is not the case because "...the relative change [in crystal number] in itself is not an indicator for the climate impact of the cirrus induced changes. Instead the climate impact depends crucially on the background optical depth of the cirrus". It is certainly true that the radiative impact of a contrail depends on multiple factors beyond crystal number alone, including the optical depth of the contrail and of the surrounding cirrus. But the effects the authors are considering here act most directly through changes in contrail crystal number. If these changes are small then changes to the contrail optical depth from these effects may naturally be expected to be small; and if significant changes in crystal number from these effects occur only rarely among significant contrails then the expectation is that significant changes in contrail optical depth from these effects will occur only rarely. The two radiation papers the authors have added citations to do not contradict these expectations in any way that I can see. Further, these and other work that I am aware of are fully consistent with the expectation I noted in my review: that the magnitude of the radiative forcing from a contrail will in general be diminished with increasing optical depth of the surrounding cirrus. That the "cirrus effects" considered in the paper can, for some specially balanced cases, happen to flip the sign of the net radiative forcing, implies no extra significance to the potential for these effects to alter the radiative impact of contrails collectively. And the authors' comment beginning "Our disagreement is caused by the fact that the reviewer assumes that the radiative forcing due to contrail perturbations is necessarily positive...." is simply false, both in the nature of our disagreement and in my assumptions about radiative forcing. Having performed numerous LES that include computations of the radiative transfer (and the feedback of radiative heating/cooling back on the contrail dynamics itself) I am well aware of the variabilities and complexities involved (see e.g., Lewellen 2014, J. Atmos. Sci. 71, 4420–4438).*

We discuss the physics relevant for estimating the climate impact in the conclusions. We say that our simulations do not allow the estimation of the climate impact as it is not the topic of our paper anyway. Therefore, we only 'speculate' about the possible climate impact in the conclusions.

Nevertheless, we appear to agree that the radiative impact of changes in cirrus properties due to contrail formation depends on the contrail induced cirrus changes and on the background cirrus properties. This means that the climate impact of the contrail induced cirrus changes depends directly on the change in the cirrus properties and on the undisturbed cirrus properties.

The remaining disagreement appears to be that the reviewer insists that from the low frequency of large changes in contrail formation caused by preexisting cirrus it can be concluded that they do not have an impact. For this argument please see our answer to A3.

*(A3) Third, they argue in comments that this is not the case "...because the places in which the impact of cirrus on contrail formation is largest are the places in which contrails can be expected to have a large impact". Lines supporting this have been added to the paper, e.g., lines 703-705 ("The change in cirrus ice crystal numbers due to contrail formation may ... have a significant influence on cirrus optical depth, radiative fluxes and cirrus life times."). That contrails can significantly impact cirrus occurrence and properties is well known. But, while this is obviously a necessary condition for "cirrus effects on contrails" to be important, it is clearly not a sufficient one. It is entirely consistent with the possibility that in the bulk of cases where contrails significantly alter existing cirrus (or create new cirrus) that the "cirrus effects on contrails" are negligible. And it is that possibility which the authors' presented results seem to me to support. If the authors continue to doubt this, I suggest they consider the following exercise: (1) estimate a cirrus IWC level required for large "cirrus effects on contrails" from their results (it seems to me about ~0.1 gm^-3); (2) estimate what fraction of all significant contrails form in conditions with cirrus IWC above that threshold.*

Thank you. We did indeed misformulate this sentence. It reads now 'The impact of cirrus ice crystals on contrail formation may, therefore, be expected to have a significant influence on cirrus optical depth, radiative fluxes and cirrus life times. (Lines 723 to 725)

In the text in the conclusions we draw on our experience of the atmospheric conditions that support contrails that have a large climate impact. Our argument is that contrails that form in cloud free air have a particularly large effect when the life times are long and that the life times are long in situations such as frontal passages and their associated conveyor belts. In exactly those situations the impact of cirrus properties on contrail formation is large which may point at those impacts having a much larger radiative effect than would be expected when considering only the frequency of occurrence.

*(A4) In places the authors try to be non-committal about the question whether "the effects that we are studying do not have an impact on contrail radiative forcing" claiming "our results do not support this conclusion, nor do they support the opposite". If the authors believe this is so, it should be stated clearly in the abstract, since this is likely to be the question of most interest to potential readers. But in fact, I think it is not so: the authors results as presented actually do support the premise of negligible impact. That this support is not entirely conclusive is, I think, due mainly to the issues in category (B).*

We do not believe that it is common practice to make statements about the topics that are not studied in the abstract.

*Category (B):*

*In simulating contrail nucleation or crystal loss, the authors neither solve the underlying physical equations involved (though such treatments exist in the literature), nor provide any estimates of the error levels in their results that may arise from the highly simplified parameterizations they use instead. The works from which they obtain their parameterizations (before modification), do employ the underlying physical equations in their development, but not for conditions that prove most important here (e.g., in the presence of cirrus with extremely high IWC). Nor do I think the original developers of these parameterizations would claim that they are (even in the absence of cirrus) accurate at the levels of precision results are quoted to throughout this paper. Such use is arguably sufficient if the authors' goal is just to speculate qualitatively about the direction of some sensitivities, or to argue that an effect is likely negligible (so that even a factor of ten error, say, would not change the basic conclusion). But that is not how the authors are presenting their results.*

*In defense of their parameterizations they offer a mix of wishful but untested speculations (e.g., "...the sensitivity of contrail ice nucleation on cirrus ice crystals is valid even if the parameterization of Kärcher et al. (2015) should generally overestimate contrail ice nucleation."; "Most of the processes (single activation time and plume inhomogeneity) may change the estimate of the overall number of contrail ice crystals but may change little in our estimate of the sensitivity of contrail ice nucleation to cirrus ice crystal sublimation/deposition within the plume."; "In summary, plume inhomogeneities and consecutive nucleation may affect the number of contrail ice crystals nucleating but may not have a significant impact on the sensitivity of contrail ice nucleation to sublimation and deposition on cirrus ice crystals", etc.) and some misleading and/or incorrect statements. Concentrating on statements in the manuscript itself, the latter include:*

As Lewellen (2020) shows the parameterization of Kärcher et al. (2015) may overestimate ice nucleation due to the lack of plume inhomogeneities and the approximation that all contrail ice crystals nucleate at the same time. In our study we calculate the increase in the plume water vapor content due to the sublimation of cirrus ice water and its impact on ice nucleation. The associated increase in relative humidity stays nearly always within the range of values found within aircraft plumes in cloud free air. This means that our application of the parameterization probes mostly the same parameter space for which the parameterization was set up. Only in very few of the situations that lead to large changes in ice nucleation would the plume experience larger relative humidity than found in plumes forming in clear air ice supersaturated areas (Those large plume supersaturations occur mainly on the level around 10.5km). This means that entrained ambient aerosols would on average have a similar impact when contrails form within cirrus than when they form in cloud free air due to the fact that fewer aerosols are entrained and relative humidity is mostly similar to the plumes developing in clear air.

*On the nucleation treatment:*

*(B1) lines 229-230: The authors are misquoting the results of Lewellen (2020) (L20 hereafter). One of the conclusions of L20 was that box-model computations of contrail nucleation in some regimes can significantly over-estimate crystal production relative to results from LES (which include much more of the correct physics). But nowhere in L20 was it stated that that problem occurred only for aerosol emissions above 10^16 (as the authors have stated in trying to argue that their simulations here are free from such problems). No fixed threshold was cited in L20 because the threshold would vary with location in the multi-dimensional parameter space. Two general regimes were identified in L20 where*

*box model results seem to reliably match LES results reasonably well, but the cases of most interest to the authors here are not in either category. The first is where nucleation is predominantly on ambient aerosol (which is not what the authors are considering here). The second is where essentially all the relevant exhaust aerosol is nucleated (the "LND" regime of L20). But for the small fraction of cases the authors are highlighting, where the presence of cirrus significantly increases the number of exhaust aerosol nucleated, this will not be the case: at the least, the comparison simulation not including the "cirrus effects" must necessarily be nucleating a significantly reduced fraction of the exhaust aerosol in order for the nucleation rate to significantly increase in the simulation with the "cirrus effects" included. In short, the simulations the authors are highlighting are precisely ones where L20 finds the box-model approach suspect. Moreover, the Kärcher et al. (2015) parameterization uses approximations (including the single activation time) above and beyond the box-model approach itself. So even restricting to where the box model reaches a target level of accuracy does not ensure that level of accuracy for the parameterization the authors are employing.*

We have modified the text saying that 'This impact is large for large aerosol emissions, e.g. for EIs = $10^{16}$ kg-fuel$^{-1}$ and higher (when using parameters as in fig. 5 of Lewellen, 2020). Added lines 230 to 231.

Our result that the increase in plume water vapor content leads to an increase in ice crystal nucleation should not depend on the approach we are using. Nevertheless, we would welcome additional studies on this topic using different methods.

*(B2) The argument alluded to in lines 230-234 and in comments (e.g., "Within existing cirrus, we can exclude entrainment of aerosols into the plume that preferentially form ice crystals.", etc.) and which apparently is used as the basis for the added lines 727-728 in the conclusions ("But, within cirrus, ambient aerosols can be expected to have a small impact on contrail formation within cirrus") does not hold. The argument seems to be that the relevant ambient aerosol would be either destroyed within the engines or already bound in cirrus ice crystals. But the elevated supersaturations encountered in the exhaust plume are such that they can easily nucleate ambient aerosol that in normal circumstances would not be nucleated in natural cirrus. And the vast majority of these aerosol are mixed into the aging plume without ever passing through the engines.*

We write that 'within existing cirrus, aerosols are not entrained that form **preferentially** ice crystals. The plume supersaturations that we see when contrails form within cirrus are similar to those in plumes forming in clear air. Only in very few of the cases in which changes in ice nucleation are large (in particular higher up in the atmosphere) does the plume experience relative humidity that is larger than the relative humidity in plumes forming in cloud free air. With lower ambient aerosol concentrations that get mixed into the plume and comparable supersaturations the problem of ambient aerosols should not be bigger in our simulations than in simulations of contrail formation within clear air. We have added lines 231 to 238 and 748 to 750.

*(B3) Regarding increased uncertainties "near-threshold": in their comments the authors try to dismiss these concerns by claiming that the example "near-threshold" LES cases included in L20 are within a "few tenths of a degree" of the threshold, while some of their cases in fig.6 with significant Delta_n/n are further away (even ~2-3 degrees below threshold and therefore not "near enough" to have elevated uncertainty). But the uncertainties in crystal production near the contrail threshold are closely related to the steep fall-off in what the authors refer to as AEI_i; as can be seen from their fig.1, this extends much further than a "few tenths of a degree" below threshold. The "near-*

*threshold" LES cases in L20 are actually 0.2 and 1.3 degrees K below threshold (depending on RH_i). The "near-threshold" uncertainty scatter illustrated in the plots there is indeed much larger for the 0.2 cases (exceeding an order of magnitude) but still large for the 1.3 K cases (tens of percent). Further, the uncertainty scatter illustrated in L20 is only that from variations in turbulence realizations. Additional sources of uncertainty, such as the deviations of the box-model results relative to LES, also grow as the threshold is approached.*

Nevertheless, we find changes more than 2 or 3K below the formation threshold that will not be strongly affected by those uncertainties. We changed the text in the conclusions slightly and say now 'Furthermore, contrail formation close to the formation threshold (within about 1K) is connected with a large uncertainty since details in the plume development may have a large impact, leading to varying contrail ice crystal numbers resulting from slightly different plume evolutions (Lewellen, 2020)'

*On the crystal loss parameterization:*

*(B4) The reviewers suggest in their comments that I have misunderstood their approach and so have expanded the discussion of it in section 2.2.4. On the contrary, I followed what they were doing the first time and the problem remains: they are not including what I would expect to be the largest potential effect of the ambient cirrus on the contrail crystal survival rate. The Kelvin-effect-dependent scavenging of moisture by large crystals from small ones is a primary component of the crystal loss in the vortex regime. It depends heavily on the size spread of crystals involved. Thine parameterization of Unterstrasser (2016) is empirically based on LES studies which include the Kelvin effect but not in the presence of ambient cirrus crystals (which are generally much larger than the contrail crystals at this stage and so can be more effective scavengers). This is true for all values of water emission in Unterstrasser's LES sets; no adjustment to the inputs into the parameterization will account for the omitted effects (including the authors' adjustment to airplane water emissions).*

We have rewritten again parts of section 2.2.4 and made hopefully even clearer that the 'adjustment to the water vapor emission' is calculated by simulating/estimating (using the diffusional growth equation) the competition of deposition / sublimation of cirrus and contrail ice crystals in the time after nucleation including the time of vortex descend. That means that we do include the impact of ice crystal sizes on deposition/sublimation considering the different sizes of cirrus and contrail ice crystals before and during vortex descend. We roughly estimate the contrail ice crystal loss after adjusting the water vapor emissions by the water that is deposited on / sublimated from cirrus ice crystals before and after vortex descend. Added lines 351 to 359.

The approximation that we make for the estimate of the sublimation of cirrus and contrail ice crystals during vortex descend is to prescribe a constant ice subsaturation instead of simulating the temporal evolution. We choose a value for the fixed ice subsaturation motivated by the simulations of Naiman et al. (2011). We now mention this simplification in the list of uncertainties of our simulations in the conclusions. The sensitivity simulation, extending the phase before vortex descent, extends the time that cirrus ice crystals can grow at the cost of contrail ice crystals and, therefore, test the sensitivity to the constant ice subsaturation approximation.

*(B5) The added clause in line 301 ("...while accounting for differences in ice crystal growth due to the Kelvin effect"), while technically true when referring to Unterstrasser (2016)'s original work, is misleading in implying that it extends to the authors' use of that parameterization here (it does not).*

This sentence is in the section in which we describe the Unterstrasser (2016) scheme without any of our modifications. A misunderstanding should therefore not be possible.

*(B6) The authors' estimate of crystal loss is not only "very rough" (line 356) but also one-sided. It includes the bulk of the cirrus effects that might aide in contrail crystal survival, but omits the one that could potentially produce the greatest additional crystal loss. Some discussion of this loss mechanism is present in the paper (added in the first revision), but the mechanism itself is not included in the authors' simulations. Thus while the authors can rightly conclude that their results support that the potential for ambient cirrus to increase contrail crystal survival in the vortex regime is negligible, more general conclusions (e.g., lines 28-29, 699-700) ruling out the potential for enhancing crystal losses are premature.*

The mechanism, the competition of deposition/sublimation dependent on ice crystal sizes is included. See answer to B4 and the description presented in section 2.2.4.

*(B7) lines 357-359, 601-604:*

*In my opinion the sensitivity study added is of negligible utility. It seems random to test a component of a parameterization of minimal importance while making no attempts to gauge uncertainties in much more important components of the parameterization.*

Since the reviewer is concerned that we do not include the competition between cirrus and contrail ice crystals, we test here the sensitivity to increasing the time period during which relative humidity is slightly supersaturated and cirrus ice crystals can grow at the cost of contrail ice crystals. Prolonging this time period leads often to more water getting deposited on cirrus ice crystals while the contrail ice crystals either grow more slowly or even decrease in size. This leads to a larger loss in contrail ice crystals.